# Higher-Order Certification for Randomized Smoothing

**Jeet Mohapatra**[1]    **Ching-Yun Ko**[1]    **Tsui-Wei Weng**[1,2]    **Pin-Yu Chen**[2]    **Sijia Liu**[2]

**Luca Daniel**[1]

[1] MIT
[2] MIT-IBM Watson AI Lab, IBM Research

## Abstract

Randomized smoothing is a recently proposed defense against adversarial attacks that has achieved state-of-the-art provable robustness against $\ell_2$ perturbations. A number of publications have extended the guarantees to other metrics, such as $\ell_1$ or $\ell_\infty$, by using different smoothing measures. Although the current framework has been shown to yield near-optimal $\ell_p$ radii, the total safety region certified by the current framework can be arbitrarily small compared to the optimal.

In this work, we propose a framework to improve the certified safety region for these smoothed classifiers without changing the underlying smoothing scheme. The theoretical contributions are as follows: 1) We generalize the certification for randomized smoothing by reformulating certified radius calculation as a nested optimization problem over a class of functions. 2) We provide a method to calculate the certified safety region using zeroth-order and first-order information for Gaussian-smoothed classifiers. We also provide a framework that generalizes the calculation for certification using higher-order information. 3) We design efficient, high-confidence estimators for the relevant statistics of the first-order information. Combining the theoretical contribution 2) and 3) allows us to certify safety region that are significantly larger than the ones provided by the current methods. On CIFAR10 and Imagenet datasets, the new regions certified by our approach achieve significant improvements on general $\ell_1$ certified radii and on the $\ell_2$ certified radii for color-space attacks ($\ell_2$ perturbation restricted to only one color/channel) while also achieving smaller improvements on the general $\ell_2$ certified radii.

As discussed in the future works section, our framework can also provide a way to circumvent the current impossibility results on achieving higher magnitudes of certified radii without requiring the use of data-dependent smoothing techniques.

## 1    Introduction

Deep neural networks (DNNs) can be highly sensitive, i.e., small imperceptible input perturbations can lead to mis-classification [1, 2]. This poses a big problem for the deployment of DNNs in safety critical applications including aircraft control systems, video surveillance and self-driving cars, which require near-zero tolerance to lack of robustness. Thus, it is important to provide guarantees for the robustness of deep neural network models against multiple worst-case perturbations. Popular threat models are the $\ell_p$-norm-based attacks, where possible perturbations are constrained in an $\ell_p$-ball with respect to a given input $x$. To that end recent research efforts have focused on attack-agnostic robustness certification which, given an input $x$, provides a *safety region* within which the model is guaranteed not to change its prediction.

*Randomized smoothing* is a recently-proposed defense [3, 4, 5] that has achieved state-of-the-art robustness guarantees. Given any classifier $f$, denoted as a *base classifier*, randomized smoothing

predicts the class that is "most likely" to be returned when noise is added to the input $x$. Thus, randomized smoothing acts as an operator that given a base classifier and a noise model, denoted as *smoothing measure*, produces a new classifier, denoted as the *smoothed classifier*. The smoothed classifiers thus produced are easily-certifiable with strong (even near-optimal for $\ell_2$) robustness guarantees under various $\ell_p$ norm threat models [3, 6, 7]).

However, for any given threat model, the state-of-the-art robustness guarantees are achieved only for the smoothed classifiers obtained using very specific smoothing measures. Thus, the classifiers that attain state-of-the-art guarantees under one threat model might perform poorly under another (Gaussian-smoothed classifiers are optimal under $\ell_2$ but perform poorly under $\ell_1$; uniform noise-smoothed classifiers are state-of-the-art under $\ell_1$ but have poor performance under $\ell_2$). Moreover, some of the recent works [8, 9, 10] show that the existing framework, which uses only the zeroth order information (the function value of the smoothed classifier $g(x)$), is incapable of producing large certified radii for $\ell_p$ norms with high values of $p$.

Motivated by the two limitations above, we focus our attention on improving the certified safety region which is agnostic of threat models. To that end, we propose a general framework to provide a larger certified safety region by better utilizing the information derived from a hard-label classifier. In particular, we summarize our contributions as follows:

1. We propose a general framework that calculates a certified safety region of a smoothed classifier $g$, around an input point $x$, by exploiting the estimated local properties (e.g. gradient, Hessian, etc.) of the classifier $g$ at $x$.

2. We give a threat-model-agnostic asymptotic-optimality result for smoothed classifiers obtained by using standard Gaussian as the smoothing measure, i.e. *gaussian-smoothed* classifiers. Using Theorem 2 in Section 3.1, we show that theoretically it is possible to produce arbitrarily tight certificates for any classifier[1]. As a consequence, we see that the impossibility results for the existing framework cannot be extended to certificates obtained using higher-order information.

3. We motivate and prove properties, like convexity (Proposition 1) and non-decreasing dependence on angle (Proposition 2), regarding the certified safety regions of gaussian-smoothed classifiers produced using the zeroth and first-order local information. Using these properties, we give formulas for calculating certified radii for gaussian-smoothed classifiers under $\ell_p$ threat models and their subspace variants with $p = 1, 2, \infty$.

4. We design new efficient estimators (see Table 1) to provide high-confidence interval estimates of relevant first-order information about the classifier since the naive Monte-Carlo estimators have prohibitively high sample complexity.

Finally, we use the 3rd and the 4th contributions above to empirically verify the effectiveness of the new certification framework in providing state-of-the-art certified accuracy for multiple threat models simultaneously. In particular, our proposed framework substantially boosts the certified accuracy for $\ell_1$ norm and subspace $\ell_2$ norm while maintaining (at times marginally improving) the state-of-the-art near-optimal results for $\ell_2$ norm. On the CIFAR10 dataset, our results for the $\ell_\infty$ norm also show improvement over the state-of-the-art bounds given by Gaussian smoothing.

## 2 Background and Related Works

### 2.1 Related Works

**Randomized Smoothing.** Randomized smoothing was initially introduced as a heuristic defense by [11] and [12]. Later, [3] formulated it as a certification method using ideas from differential privacy, which was then improved by [6] using Renyi divergence. For gaussian-smoothed classifiers, [5] made the certified bounds worst-case-optimal in the context of certified $\ell_2$ norm radii by using the Neyman-Pearson Lemma, while authors of [13] combined the certification method with adversarial training to further improve the empirical results. Along another line of works, some extended existing certification methods to get better $\ell_p$ norm certified radii using different smoothing distributions, e.g.

a discrete distribution for $\ell_0$ certificates [14], the Laplace distribution for $\ell_1$ certificates [15], and the generalized Gaussian distribution for $\ell_\infty$ certificates [16]. Recently, [8] proposed a general method for finding the optimal smoothing distribution given any threat model, as well as a framework for calculating the certified robustness for the smoothed classifier.

**Impossibility Results.** Recently, a number of works have shown that for $\ell_p$ norm threat models with large $p$, it is impossible to give a big certified radius $\left(O(d^{\frac{1}{p}-\frac{1}{2}})\right.$ where $d$ is the input dimension$)$ while retaining a high standard accuracy. In particular, the results on $\ell_\infty$ threat model given in [9, 10] and the results on $\ell_p$ (for sufficiently large $p$) threat models given in [8] establish a certification/accuracy trade-off, which also exaggerates the need for an extended and generalized framework that breaks the confined trade-off and impossibility results.

## 2.2 Preliminaries and Notations

Generally, a classifier in machine learning is represented as a function $f$ that maps from the feature space $\mathbb{R}^d$ to a probability vector over all the classes $\mathcal{Y}$. We use $\mathcal{F}$ to denote the set of *all* classifiers $f : \mathbb{R}^d \mapsto \Delta^{\mathcal{Y}}$, where $\Delta^{\mathcal{Y}}$ is a probability simplex collecting all possible probability vectors with dimension $\mathcal{Y}$. In this paper, we work under the restrictive black-box model, i.e., we assume we only have access to the class labels outputted by classifier $f$. For ease of use, we consider the output of $f$ to be the hard-thresholding of the probability vector, i.e. the output is a one-hot encoding vector. Given a black-box classifier $f \in \mathcal{F}$, randomized smoothing is a technique that creates a smoothed classifier $g$ by convolving $f$ (hereafter referred to as the *base classifier*) with a *smoothing measure* $\mu$, i.e., $g = f \star \mu$. We use $\mathcal{G}_{\mathcal{F}}^{\mu}$ to denote the class of these smoothed functions and $\star$ denotes the convolution operator. Note that the smoothing measure $\mu$ is a probability measure and each smoothed function $g \in \mathcal{G}_{\mathcal{F}}^{\mu}$ defines a mapping from $\mathbb{R}^d$ to $\Delta^{\mathcal{Y}}$. Thus, we have $\mathcal{G}_{\mathcal{F}}^{\mu} \subset \mathcal{F}$.

## 3 A General Framework for Randomized Smoothing

Given a smoothing measure $\mu$, the certified safety region around a point $x$ is given as the subset of the space over which the classifier output is guaranteed not to change. In practice, this involves estimating some local information about $g$ around the point $x$ and giving the certified safety region as the common subset (intersection) of the safety region of all classifier $h \in \mathcal{G}_{\mathcal{F}}^{\mu}$ whose local properties match the estimated local information. In the current literature, the only local information used is the function value $g(x)$ [5, 13]; however, more local information of $g(x)$, such as the gradient or higher-order derivatives, can in fact be exploited for deriving better certified radii. To illustrate the idea, we express the local information as the local constraints, and re-write the problem of finding the certified radius in the following: Let $H_i^x(g)$ be an estimate of some local property of $g$ at the input point $x$, and let there be $k$ such constraints, then the certified safety region $\mathbf{SR}(x)$ can be written as

$$\mathbf{SR}(x) = \bigcap \left\{ \mathbf{S}^h(x) \;\middle|\; h \in \mathcal{G}_{\mathcal{F}}^{\mu} \text{ and } H_i^x(h) = H_i^x(g),\; 1 \le i \le k \right\},$$
$$\mathbf{S}^h(x) = \left\{ \delta \;\middle|\; \arg\max h(x + \delta) = \arg\max h(x) \right\} \tag{1}$$

where $\mathbf{S}^h(x)$ gives the safety region for the function $h$. The existing state-of-the-art certificates [5] via randomized smoothing are a special case of this framework with $k = 1$ and $H_1^x(h) = (h(x))_A$ given $H_1^x(g) \ge p_A$, where $h(\cdot)_A$ denotes the $A^{th}$ component of the vector $h(\cdot)$ and $p_A$ is the lower bound of the estimated probability that $x$ belongs to the predicted class $A$.

**Reduction to Binary.** In existing literature, certificates for randomized smoothing based classifiers effectively reduce the multi-class classification to binary by reducing the problem to *top-1 prediction vs other classes*. We follow the same idea to simplify our certification objectives by observing :

$$h(x + \delta)_{\arg\max(h(x))} > 0.5 \implies \arg\max(h(x + \delta)) = \arg\max(h(x)).$$

Assuming $\arg\max(h(x)) = c$, we give by $\mathbf{S}_L^h(x) = \{\delta \mid h(x + \delta)_c > 0.5\}$ a set lower bound of $\mathbf{S}^h(x)$, i.e., $\mathbf{S}_L^h(x) \subset \mathbf{S}^h(x)$. Substituting $\mathbf{S}^h(x)$ in (1) by $\mathbf{S}_L^h(x)$ gives us a set lower bound $\mathbf{SR}_L(x)$ of $\mathbf{SR}(x)$ with the safety region subset:

$$\mathbf{SR}_L(x) = \bigcap \left\{ \{\delta \mid h(x + \delta)_c > 0.5\} \;\middle|\; h \in \mathcal{G}_{\mathcal{F}}^{\mu} \;\&\; H_i^x(h) = H_i^x(g),\; 1 \le i \le k \right\}$$

This set can be re-written as $\mathbf{SR}_L(x) = \{\delta \mid \mathbf{p}_x(x + \delta) > 0.5\}$, where the probability function $\mathbf{p}_x(z)$ given by the optimization problem:

$$\mathbf{p}_x(z) = \min_{h \in \mathcal{G}_{\mathcal{F}}^{\mu}} h(z)_c \quad \textbf{s.t.} \quad H_i^x(h) = H_i^x(g),\ 1 \leq i \leq k. \tag{2}$$

**Our proposed framework.** Use Generalized Neymann Pearson Lemma [17] to solve Equation (2) for $\mathbf{p}_x(z)$. The safety region is then given as super-level set of $\mathbf{p}_x$, i.e., $\mathbf{SR}_L(x) = \{\delta \mid \mathbf{p}_x(x+\delta) > 0.5\}$.

**Discussion.** Under the proposed general framework for calculating certified radii, it is easy to see that adding more local constraints ($H_i^x$) in Equation (2) gives a bigger value of $\mathbf{p}_x(z)$ for any $x, z$ which makes the super-level set of $\mathbf{p}_x$, equivalently the certified safety region, bigger. In the following subsection, we study the properties of functions in $\mathcal{G}_{\mathcal{F}}^{\mu}$ to get some motivation about which information might help us achieve a larger certified safety region. Then, we provide an example usage of this framework that exploits additional local higher-order information to give larger certified radii. The proofs of all theoretical results hereafter are supplemented in the appendix.

### 3.1 Regularity Properties of $\mathcal{G}_{\mathcal{F}}^{\mu}$

For any black-box classifier $f$, the function map between the input pixel-space $\mathbb{R}^d$ and the prediction $\Delta^{\mathcal{Y}}$ is discontinuous. As a result, the higher-order derivatives might not always exist for the smoothed function $g$ for smoothing with general probability measures $\mu$. Therefore, it is crucial to establish conditions on the probability measure $\mu$ that guarantee the existence of higher-order derivatives for $g$ at all points in the pixel-space $\mathbb{R}^d$. We give this in the following theorem:

**Theorem 1.** *If $\forall \alpha \in \mathbb{N}^d$, $\int_{\mathbb{R}^d} |D_x^{\alpha} \mu(y - x)| dy$ [2] is finite, then $\mathcal{G}_{\mathcal{F}}^{\mu} \subset \mathcal{C}^{\infty}$. Moreover, if $g \in \mathcal{G}_{\mathcal{F}}^{\mu}$ is given as $g = f \star \mu$ for some $f \in \mathcal{F}$, then*

$$\nabla^i g(x) = \int_{\mathbb{R}^d} f(y)(-1)^i \nabla^i \mu(y - x) dy.$$

**Corollary 1.1.** *When $\mu$ is the isotropic Gaussian distribution $\mathcal{N}(0, \sigma^2 \mathcal{I})$, then $\mathcal{G}_{\mathcal{F}}^{\mu} \subset \mathcal{C}^{\infty}$ and $\nabla^i g(x) = \int_{\mathbb{R}^d} f(y)(-1)^i \nabla^i \mu(y - x) dy$.*

Corollary 1.1 guarantees the existence of higher-order derivatives for functions obtained by Gaussian smoothing and also provides a way of evaluating the higher-order terms using only the output of the base classifier $f$. Considering the fact that the truncated $m^{th}$ order Taylor expansion around $x$ gives better local approximations of the function with an increasing $m$, we expect to get better approximations of the safety region by using the higher-order information about $g$ at $x$. Now in order to understand the limitations of this technique we consider the asymptotic limit where we have all the higher order terms of $g$ at $x$. We observe that:

**Theorem 2.** *When $\mu$ is the isotropic Gaussian distribution $\mathcal{N}(0, \sigma^2 \mathcal{I})$, then $\forall g \in \mathcal{G}_{\mathcal{F}}^{\mu}$, $g$ is a real analytic function with infinite radius of convergence, i.e., the Taylor series of $g$ around any point $w$ converges to the function $g$ everywhere.*

**Asymptotic-Optimality Remark:** As a consequence of Theorem 2, when $\mu$ is the isotropic Gaussian distribution, any function $h \in \mathcal{G}_{\mathcal{F}}^{\mu}$ is uniquely identified by its Taylor expansion. Thus if we can accurately estimate all the higher-order information about $g$ at any point $x$, along with the fact that $g \in \mathcal{G}_{\mathcal{F}}^{\mu}$, the feasible set in problem (2) will reduce to the singleton set containing only the function $g$ making $\mathbf{p}_x(z) = g(z)_c$. This gives us the safety region $\mathbf{SR}_L(x) = \{g(x)_c > 0.5\}$ which is the exact safety region under the binary, top-1 vs rest, relaxation (abstain when top-1 class has probability less than 0.5). Computing the exact safety region implies that we can get large certified $\ell_p$ radii for all $p$. Hence, the impossibility results shown in the recent works by Yang et.al [8], Blum et al. [9] and Kumar et al. [10] or any of their extensions can not hold for certification methods that use higher-order information. See APPENDIX C for remarks on optimality of certificates obtained using only zeroth and first order information.

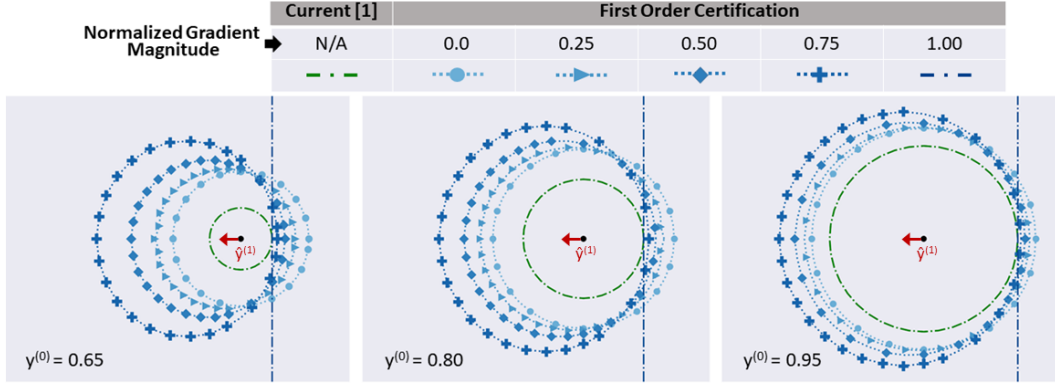

Figure 1: A longitudinal slice of the certified safety region around a point under various values of $y^{(0)}, y^{(1)}$ keeping direction of $y^{(1)}$ fixed along the negative x axis. Normalized gradient magnitude is given as $\|y^{(1)}\|_2 / \max_{g(x)=y^{(0)}} \|\nabla g(x)\|_2$. The green region is certified using only zeroth order information and the blue regions using both zeroth and first order information.

## 3.2 Certification For Randomized Smoothing Using First-Order Information

For the rest of this paper, we focus on the certified region produced by solving the optimization problem described in Equation (2) using the zeroth and first-order local information of $g$ at a point $x$. The resulting first-order optimization problem is given as follows:

$$\mathbf{p}_x(z) = \min_{h \in \mathcal{G}_{\mathcal{F}}^{\mu}} h(z)_c \quad \textbf{s.t.} \quad h(x) = y^{(0)}, \nabla h(x) = y^{(1)} \tag{3}$$

where $y^{(0)}, y^{(1)}$ are the zeroth and first-order local information of $g$ at $x$ estimated using Corollary 1.1. In practice, it is only possible to give interval estimates for $y^{(0)}, y^{(1)}$. The following Theorem gives a lower bound for $\mathbf{p}_x$ given interval estimates of $y^{(0)}, y^{(1)}$:

**Theorem 3** (Lower Bound of $\mathbf{p}_x(z)$). *For a base classifier $f \in \mathcal{F}$, if $g = f \star \mu$, $\mu$ is the isotropic Gaussian distribution $\mathcal{N}(0, \sigma^2 \mathcal{I})$, $y^{(0)} = g(x)$, $y^{(1)} = \nabla g(x)$, then for any unit vector $v$ and any positive value of $r$, $\mathbf{p}_x(x + \sigma r v)$ can be lower bounded by solving the following set of equations:*

$$\int_{-\infty}^{\infty} \frac{1}{\sqrt{2\pi}} e^{-\frac{x^2}{2}} \Phi(c(x)) dx = q \tag{4}$$

$$\int_{-\infty}^{\infty} \frac{1}{\sqrt{2\pi}} e^{-\frac{x^2}{2}} \frac{1}{\sqrt{2\pi}} e^{-\frac{c(x)^2}{2}} dx = m_2 \quad \text{(5a)} \qquad \int_{-\infty}^{\infty} \frac{1}{\sqrt{2\pi}} x e^{-\frac{x^2}{2}} \Phi(c(x)) dx = m_1 \quad \text{(5b)}$$

*with $q \le y^{(0)}, m_1 \le \sigma v^T y^{(1)}, m_2 \le \sigma \|y^{(1)} - v^T y^{(1)} v\|_2$, $c(x) := c_0 + c_1 x + c_2 e^{rx}$, and $\Phi(z)$ being the CDF of the standard normal distribution. If the solution $(c_0, c_1, c_2)$ of above equations has $c_1 < 0$, then the lower bound of $\mathbf{p}_x(x + \sigma r v)$ is instead given by solving Equations (4) to (5a) with $c(x) := c_0 + c_2 e^{rx}$.*

Given $\mathbf{p}_x(z)$ the certified safety region can be given as the super-level set $\{z \mid \mathbf{p}_x(z) > 0.5\}$. To supplement the above theorem, we point out that using only Equation (4) gives exactly the zeroth-order certification. Intuitively, we see that unlike the zeroth-order certification that is calculated through equations independent of $v$, first-order certification uses additional Equations (5a), (5b) that depend on $v$, breaking the symmetry and admitting non-isotropic certified radius bounds. Referring to Theorem 3, we observe that $\mathbf{p}_x(x + rv)$ only depends on $g(x), \|\nabla g(x)\|_2$ and angle between $v$ and $\nabla g(x)$. Thus, the first order certified region has a cylindrical symmetry with the axis along the vector $y^{(1)}$.

In Figure 1, we compare a longitudinal slice of the safety region around a point $x$ calculated using both zeroth and first-order information to the longitudinal slice of the safety region obtained using only the zeroth-order information. We use these figures to deduce some properties of these safety regions:

**i)** The relative improvement in the size of the safety region is the largest for low values of $y^{(0)}$ and get successively smaller with an increasing $y^{(0)}$.

**ii)** Given a fixed value of $y^{(0)}$, the volume of the certified safety region grows to infinity (the half-space given by the dash line) as the magnitude of $\left\|y^{(1)}\right\|_2$ increases to its maximum possible value.

**iii)** The certified safety region is convex and for any given values of $y^{(0)}$ and $\left\|y^{(1)}\right\|_2$, the directional certified radius is highest along the direction of the $y^{(1)}$ and gets successively lower as we rotate away from it (the angle between $v$ and $y^{(1)}$ increases), to the lowest in the direction opposite to the $y^{(1)}$.

We formalize the third observation into the following propositions, which allows us to calculate the certified radii for the various threat models discussed in the rest of this section.

**Proposition 1.** *The certified safety region, $\mathbf{SR}_L(x)$, calculated using the zeroth and first-order local information is convex, i.e., if $x_1, x_2 \in \mathbf{SR}_L(x)$ then $\frac{x_1+x_2}{2} \in \mathbf{SR}_L(x)$.*

Using Prop 1, we see that along any vector $v$ if for some $R$, $x + Rv \in \mathbf{SR}_L(x)$, then forall $0 \le r \le R$, $x + rv \in \mathbf{SR}_L(x)$. Thus, along any direction $v$, we can define a directional robustness $\mathbf{R}_v$ such that forall $0 \le r < \mathbf{R}_v$, $x + rv \in \mathbf{SR}_L(x)$ and forall $r \ge R$, $x + rv \notin \mathbf{SR}_L(x)$. It is easy to see that $\mathbf{R}_v$ can be given as the solution to the equation $r \ge 0$, $\mathbf{p}_x(x + rv) = 0.5$.

**Proposition 2.** *For any given value of $y^{(0)}, y^{(1)}$, the directional robustness along $v$, $\mathbf{R}_v$, given by the first-order certification method is a non-increasing function of the angle between $v$ and $y^{(1)}$, i.e., $\cos^{-1}\left(\frac{v^T y^{(1)}}{\|v\|_2 \|y^{(1)}\|_2}\right)$.*

**(I) Certified $\ell_2$ Norm Radius:** By Proposition 2, we have that for first-order certification, the directional robustness along all directions $v$ is at least as big as the directional robustness along the direction $-y^{(1)}$, implying that the $\ell_2$ norm certified radius is equal to $\mathbf{R}_{-y^{(1)}}$. Thus, we have

**Corollary 3.1** (Certified $\ell_2$ Norm Radius). *For a base classifier $f \in \mathcal{F}$, if $g = f \star \mu$, where $\mu$ is the isotropic Gaussian distribution $\mathcal{N}(0, \sigma^2 \mathcal{I})$, $y^{(0)} = g(x)$, $y^{(1)} = \nabla g(x)$, the $\ell_2$ norm radius $R$ is given as $R = \sigma r$, where $(r, w_1, w_2)$ is the solution of the system of equations:*

$$\Phi(w_1 - r) - \Phi(w_2 - r) = 0.5 \tag{6}$$

$$\Phi(w_1) - \Phi(w_2) = q \tag{7a} \qquad \frac{1}{\sqrt{2\pi}}e^{-\frac{w_2^2}{2}} - \frac{1}{\sqrt{2\pi}}e^{-\frac{w_1^2}{2}} = m_1 \tag{7b}$$

*with $q \le y^{(0)}$ and $m_1 \ge \sigma\left\|y^{(1)}\right\|_2$.*

Corollary 3.1 gives the same result as Cohen et al. [5] if we remove constraint (7b) and consider the minimum value of $r$ produced by Equation (6) over all possible pairs of $w_1, w_2$ satisfying equation (7a) . As a result the certified $\ell_2$ norm radius given by Corollary 3.1 is always greater than or equal to the radius given by the existing framework with equality holding only when the classifier is linear.

**(II) Certified $\ell_1, \ell_\infty$ Norm Radius:** Using Proposition 1 we see that the minimum $\ell_1$ norm radius is along one of the basis vectors and using Proposition 2 we see that the minimum directional robustness must be along the basis vector with the biggest angle with $y^{(1)}$. Thus, the projection of $y^{(1)}$ along this can be given as $-\left\|y^{(1)}\right\|_\infty$.

**Corollary 3.2** (Certified $\ell_1$ Norm Radius). *For a base classifier $f \in \mathcal{F}$, if $g = f \star \mu$, where $\mu$ is the isotropic Gaussian distribution $\mathcal{N}(0, \sigma^2 \mathcal{I})$, $y^{(0)} = g(x)$, $y^{(1)} = \nabla g(x)$, the $\ell_1$ norm radius $R$ is obtained by solving $\mathbf{p}_x(x + Rv) = 0.5$, where $\mathbf{p}_x(x + Rv)$ is given by solving the problem in Theorem 3 with $m_1 \le -\sigma\left\|y^{(1)}\right\|_\infty$, $m_2 \le \sigma\sqrt{\left\|y^{(1)}\right\|_2^2 - \left\|y^{(1)}\right\|_\infty^2}$.*

Similarly, the minimum $\ell_\infty$ norm radius must ocuur along a vector of all 1's and -1's and using Proposition 2 we see it must be along the vector of all 1's and -1's with the biggest angle with $y^{(1)}$. Thus, the projection of $y^{(1)}$ along this can be given as $-\left\|y^{(1)}\right\|_1$.

**Corollary 3.3** (Certified $\ell_\infty$ Norm Radius). *For a base classifier $f \in \mathcal{F}$, if $g = f \star \mu$, where $\mu$ is the isotropic Gaussian distribution $\mathcal{N}(0, \sigma^2 \mathcal{I})$, $y^{(0)} = g(x)$, $y^{(1)} = \nabla g(x)$, the $\ell_\infty$ norm radius $R$ is obtained by solving $\mathbf{p}_x(x + Rv) = 0.5$, where $\mathbf{p}_x(x + Rv)$ is given by solving the problem in Theorem 3 with $m_1 \le -\frac{\sigma}{\sqrt{d}}\left\|y^{(1)}\right\|_1$, $m_2 \le \frac{\sigma}{\sqrt{d}}\sqrt{d\left\|y^{(1)}\right\|_2^2 - \left\|y^{(1)}\right\|_1^2}$.*

Table 1: Estimators to calculate the different norm value of $\nabla g(x)$. (* newly designed estimators)

| | $\ell_1$ | $\ell_2$ | $\ell_\infty$ |
|---|---|---|---|
| constants | $t = \sqrt{\frac{2kd(d\log 2 - \log\alpha)}{n_1+n_2}}$ | $\epsilon_u = \sqrt{\frac{-k(n_1+n_2)\log\frac{\alpha}{2}}{2n_1 n_2(X_{n_1}^T Y_{n_2}+t)}}$ $t = \sqrt{-k^2\frac{\sqrt{2}d}{n_1 n_2}\log\frac{\alpha}{2}}$ $\epsilon_l = \sqrt{\frac{-k(n_1+n_2)\log\frac{\alpha}{2}}{2n_1 n_2(X_{n_1}^T Y_{n_2}-t)}}$ | $t = \sqrt{\frac{2k(\log 2d - \log\alpha)}{n_1+n_2}}$ |
| upper bound | $\left\|\frac{n_1 X_{n_1}+n_2 Y_{n_2}}{n_1+n_2}\right\|_1 + t$ | $\frac{\sqrt{X_{n_1}^T Y_{n_2}+t}}{\sqrt{1+\epsilon_u^2}-\epsilon_u}$ (*) | $\left\|\frac{n_1 X_{n_1}+n_2 Y_{n_2}}{n_1+n_2}\right\|_\infty + t$ |
| lower bound | $\left\|\frac{n_1 X_{n_1}+n_2 Y_{n_2}}{n_1+n_2}\right\|_1 - t$ | $\frac{\sqrt{X_{n_1}^T Y_{n_2}-t}}{\sqrt{1+\epsilon_l^2}+\epsilon_l}$ (*) | $\left\|\frac{n_1 X_{n_1}+n_2 Y_{n_2}}{n_1+n_2}\right\|_\infty - t$ |

**(IV) Certified $\ell_p$ Norm Radius over a Subspace:**

**Definition 1** (Subspace Certified $\ell_p$ Norm Radius). *Given subspace S, we define the $\ell_p$ certified radius in the subspace as follows:*

$$\max_R R \quad \textbf{s.t.} \ \forall\, \delta \in S, \|\delta\|_p \le R; \ \arg\max(g(x+\delta)) = \arg\max(g(x)).$$

Going beyond $\ell_p$ norm threat models, we give the subspace $\ell_p$ norm threat model that allows us to, among other things, measure the sensitivity of a deep learning model over a subset of the pixels instead of the entire image. Using Propositions 1 and 2, we are able to extend our results to give :

**Corollary 3.4** (Subspace Certified $\ell_p$ norm radius). *For a base classifier $f \in \mathcal{F}$, if $g = f \star \mu$, where $\mu$ is the isotropic Gaussian distribution $\mathcal{N}(0, \sigma^2 \mathcal{I})$, $y^{(0)} = g(x)$, $y^{(1)} = \nabla g(x)$, and a subspace $S$ with orthogonal projection matrix $P_S$, for $p = 1, 2, \infty$ the subspace $\ell_p$ norm certified radius $R$ is obtained by solving $\boldsymbol{p}_x(x+Rv) = 0.5$, where $\boldsymbol{p}_x(x+Rv)$ is given by solving the problem in Theorem 3 with $m_1 \le -\sigma\left\|P_S y^{(1)}\right\|_{p'}, m_2 \le \sigma\sqrt{\left\|y^{(1)}\right\|_2^2 - \left\|P_S y^{(1)}\right\|_{p'}^2}$, and $\|\cdot\|_{p'}$ is the dual norm of $\|\cdot\|_p$.*

## 4 Numerical Estimation of First-Order Information

In the previous section a concrete set of methods has been proposed to calculate the first-order certificate for randomized smoothing assuming we have access to the first-order information $y^{(1)}$. Therefore in this section we complete the framework by giving ways to get high confidence estimates of vector $y^{(1)}$. Recalling that the hard-label classifier $f$ is not continuous and hence not differentiable, we need to use the zeroth order information about $f$ to approximate $y^{(1)}$ , i.e., use the result from Corollary 1.1 : $y^{(1)} = \mathbb{E}_{y\sim\mu}[\frac{y}{\sigma^2}f(x+y)]$. As the estimator $\frac{y}{\sigma^2}f(x+y)$ is noisy [13], approximating the expectation using Monte-Carlo sampling is hard, i.e., the number of samples required to achieve a non-trivial high-confidence estimate of $y^{(1)}$ scales linearly with the input dimension. In practice, this translates to a sample complexity of 10 billion for CIFAR and near a trillion for Imagenet.

Although estimating the vector $y^{(1)}$ is hard, we observe from Corollary 3.2, 3.1, 3.3, 3.4 that we don't need the whole vector $y^{(1)}$ but only $\left\|y^{(1)}\right\|_2, \left\|y^{(1)}\right\|_\infty$ for $\ell_1$ certified radius; $\left\|y^{(1)}\right\|_2$ for $\ell_2$ certified radius; $\left\|y^{(1)}\right\|_1, \left\|y^{(1)}\right\|_2$ for $\ell_\infty$ certified radius; and $\left\|P_S y^{(1)}\right\|_p, \left\|y^{(1)}\right\|_2$ for the subspace $\ell_p$ certified radius ($p = 1, 2, \infty$). Most of these statistics can be estimated much more efficiently due to the following observation:

**Theorem 4.** *Given a black-box classifier $f$ and the random vector $z = w(f(x+w)_c - \frac{1}{2})$ where $w \sim \mathcal{N}(0, \sigma^2\mathbf{I})$, we have that $z - \sigma^2 y^{(1)}$ is a sub-gaussian random vector with parameter $k = \sigma^2(\frac{1}{4} + \frac{3}{\sqrt{8\pi e}})$. For convenience, we do some abuse of notation to denote this as $(z - \sigma^2 y^{(1)}) \sim \mathrm{subG}(k)$.*

Leveraging the properties of sub-gaussian random variables, we establish the following general result:

**Theorem 5.** *For any $\alpha \ge 2e^{-\frac{d}{16}}$, if we have two random vectors $X, Y$ such that $(X - \beta) \sim \mathrm{subG}(k_1)$ and $(Y - \beta) \sim \mathrm{subG}(k_2)$, then we can show that using $t = \sqrt{-\sqrt{2}k_1 k_2 d \log\frac{\alpha}{2}}$, $\epsilon_u =$*

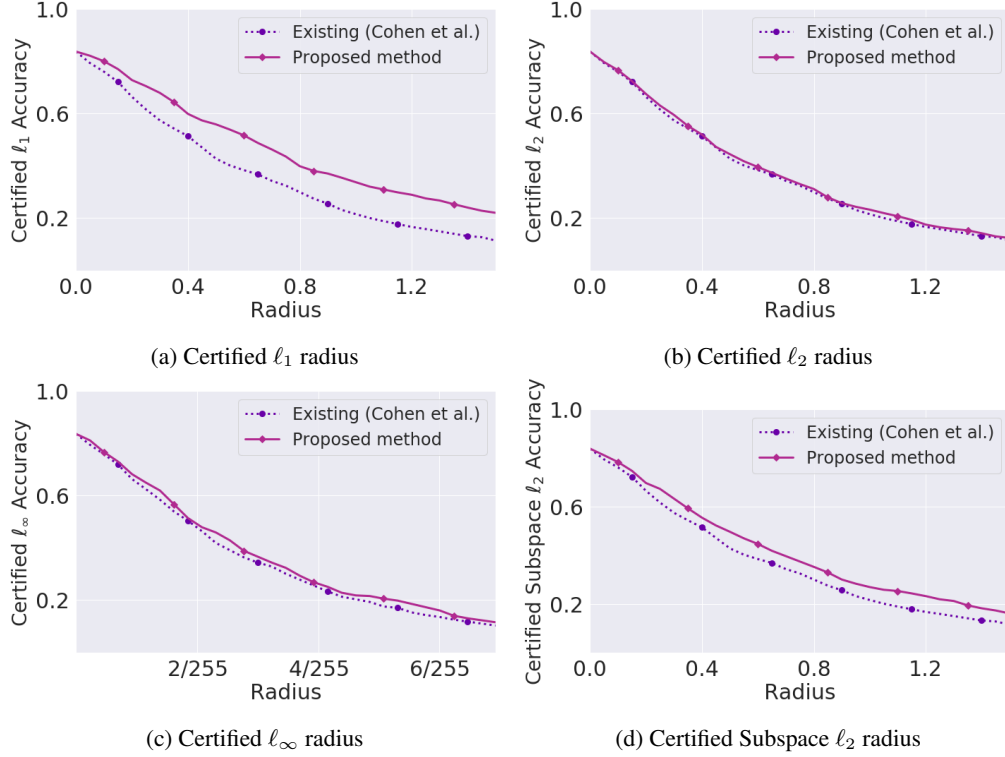

(a) Certified $\ell_1$ radius

(b) Certified $\ell_2$ radius

(c) Certified $\ell_\infty$ radius

(d) Certified Subspace $\ell_2$ radius

Figure 2: Increase in Certified Accuracy for CIFAR10 seen under various threat models

$$\sqrt{\frac{-(k_1+k_2)\log\frac{\alpha}{2}}{2(X^TY+t)}}, \text{ and } \epsilon_l = \sqrt{\frac{-(k_1+k_2)\log\frac{\alpha}{2}}{2(X^TY-t)}} \text{ renders}$$

$$\mathbb{P}\left(\|\beta\|_2 \leq \frac{\sqrt{X^TY+t}}{\sqrt{1+\epsilon_u^2}-\epsilon_u}\right) \geq 1-\alpha, \quad \mathbb{P}\left(\|\beta\|_2 \geq \frac{\sqrt{X^TY-t}}{\sqrt{1+\epsilon_l^2}+\epsilon_l}\right) \geq 1-\alpha.$$

Now, let $X = X_{n_1}, Y = Y_{n_2}$ be the empirical average of $n_1, n_2$ independent samples of the random variable $z$. Then, using Theorem 4 and 5 allows us to give the estimators illustrated in Table 1. However, we see that estimation of $\|y^{(1)}\|_1$ still scales linearly with $d$ making it impractical. Thus, the $\ell_\infty$ norm certified radius is still given as $\frac{1}{\sqrt{d}}$ of the $\ell_2$ norm certified radius.

Finally, for estimating the $\ell_p$ norm ($p = 1, 2, \infty$) over a subspace $S$ we show that we can use the corresponding estimators from Table 1 except with $X_{n_1}^S = P_S X_{n_1}, Y_{n_2}^S = P_S Y_{n_2}$ and $d = d_S$ where $P_S$ is the orthogonal projection matrix for $S$ and $d_S$ is the dimension of $S$.

## 5   Experimental Results

We empirically study the performance of the new certification schemes on standard image classification datasets, CIFAR10 and Imagenet. We reuse the models given by Cohen et al. [5] and calculate the certified accuracy at radius $R$ by counting the samples of the test set that are correctly classified by the smoothed classifier $g$ with certified radii of at least $R$. For both our proposed certificate and the baseline certificate [5], we use a failure probability of $\alpha = 0.001$ and $N = 200,000$ samples for CIFAR10 and $N = 1,250,000$ samples for Imagenet. For $\ell_\infty$ radius we require a lot more samples to get better results as our current estimator is too noisy. In order to show the possibility presented by our approach, we also provide results for the $\ell_\infty$ certified radius for CIFAR10 estimated using $4,000,000$ samples. In our plots, we present, for each threat model the upper envelopes of certified accuracies attained over the range of considered $\sigma \in \{0.12, 0.25, 0.50, 1.00\}$. Further experiments (including all Imagenet experiments) are given in the appendix.

## 5.1 Certified Accuracy

As expected we see from Figure 2b that the new framework gives only marginal improvement over the $\ell_2$ radius certified by existing methods. This follows from the fact that the existing methods already produce near-optimal certified $\ell_2$ radii. However, certifying a significantly bigger certified safety region allows us to give significant improvements over the certified $\ell_1$ radius, the $\ell_\infty$ radius and the subspace $\ell_2$ radii (the subspace considered here is the red channel of the image, i.e., we only allow perturbations over red component of the RGB pixels of the image). We note here that considering the performance for individual threat models, the $\ell_1$ norm certified accuracy given here is still smaller than the existing state-of-the-art. However, the current state-of-the-art for $\ell_1$ certified radius uses the uniform smoothing distribution which performs quite poorly for $\ell_2$ norm radius. If we consider multiple threat models simultaneously the proposed method gives the best joint performance. Similar findings are also reported for Imagenet experiments (given in the appendix).

Moreover, we see that for the certified $\ell_\infty$ radius the new method gives an improvement over the certified $\ell_\infty$ radius provided by existing methods for Gaussian noise. As Gaussian noise was shown to achieve the best possible certified $\ell_\infty$ radius for existing methods, this provides an improvement over the best possible $\ell_\infty$ radius certifiable using existing methods. Although the improvements are small in magnitude, we believe they can be improved upon by using tighter estimators for $\left\|y^{(1)}\right\|_1$.

## 6 Conclusion and Future Work

In this work, we give a new direction for improving the robustness certificates for randomized-smoothing-based classifiers. We have shown that even in the black-box (hard-label classifier) setting, leveraging more information about the distribution of the labels among the sampled points allows us to certify larger regions and thus guarantee large certified radii against multiple threat models simultaneously. We have shown this to hold theoretically and also demonstrated it on CIFAR and Imagenet classifiers. Additionally, for gaussian smoothing, the proposed framework gives a way to circumvent the recently-proposed impossibility results and also promises a threat-model-agnostic asymptotic-optimality result. However, we note that the first-order smoothing technique given in this paper is only a proof-of-concept to show it is possible to better leverage local information to certify larger safety regions without changing the smoothing measure. In future, this work could be extended to derive and use, for any given threat model, the best local information to exploit in order to improve the certificates for that threat model.

## Broader Impact

In recent years, machine learning and intelligent systems have started to be widely adopted into everyday life, including several safety-critical applications. The wide-spread use of these systems requires them to be held to a higher level of scrutiny. One such requirement is "robustness", i.e., the systems' prediction should not be too sensitive to small input perturbations. In general, undefended/vanilla deep learning models have been found to be extremely sensitive to small imperceptible perturbations. This in turn has given rise to many defense techniques, able to overcome "existing" threats to robustness. However most of these models have later proved to be ineffective against newer threats. In turn, this has then given rise to a class of techniques which provide some mathematical guarantees about the robustness of their predictions. In this paper, we extended one such framework that provides some mathematically provable guarantees against any "single individual" threat model chosen from a large class of threat models. Specifically, we provided a way to construct models with some mathematically provable guarantees against "multiple simultaneous" threat models.

The *benefits* of this contribution include providing a more holistic picture of the robustness guarantees for deployed models. This can hopefully bring us a step closer to trustworthy AI. Since this work helps building models that give some guarantees on their behavior, we hope it would also lead to the wider adoption of deep learning models. Moreover, we hope that the guarantees given by these models would allow people to have some improved sense of security when using deep-learning-based products. Considering intensive applications such as product matching, categorization in assembly lines or video surveillance that require long periods of hard effort, robust models will give more reliable and efficient means of achieving the task and considerably reducing the burden on humans.

As for the potentially *negative* effects of this contribution, we feel that any progress towards robustness certification could easily become a double-edged sword. This is because, if adopted blindly (i.e. specifically without extremely careful attention to the types of guarantees provided and most importantly those NOT provided), it may give also the **false** sense of security that current deep-learning systems are already ready for deployment. These robustness concerns are one of the major bottlenecks for the adoption of deep-learning models in safety-critical applications such as self-driving car and air traffic control systems. hence the ability to provide "some" robustness guarantees might result in wide premature adoption of deep learning models in such applications. We would like to candidly admit to developer and user readers of this paper that the presence of some robustness guarantees for a model does not mean we understand the model or that it is completely safe to deploy it. To the contrary the authors of this paper believe that we are still quite far from such safe adoption scenario. Further issues lie also beyond robustness. A negative effect of this publication and publication like this on robustness, is might give the highly incorrect impression that with robustness one can feel safe in deploying AI systems in society. On the contrary we would also like to candidly admit and remind the readers that deep-learning-based models suffer from problems such as lack of accuracy guarantees for out-of-distribution data , lack of fairness, lack of explain-ability and many others that MUST to be solved before AI systems are viable for real-world applications. More specifically about the lack of accuracy guarantees for out-of-distribution data: the robustness of a model does not necessarily mean its prediction is always accurate. Robustness and accuracy are indeed two disjoint concepts. While it is well-known that an accurate prediction might not be robust, it is also essential to keep in mind that a robust prediction need not always be accurate. In other words, some models may generate a prediction that is indeed highly robust (i.e. does not change upon perturbation) but consistently incorrect! As a result, in applications such as air traffic control systems, the models might display extremely bad behaviour because of situations not present in its training, while the presence of some robustness guarantees might give a false sense of security to an inexperience user (e.g. not familiar with the admittedly subtle mathematical intricacies of different threat models), that these systems are completely fault-tolerant.

In conclusion, we would like to remark that although this paper helps to take a step towards building more trustworthy AI, that goal is indeed still quite far-off.

## Funding Disclosure

This work was funded by the MIT-IBM collaboration under the research grant " Toward Trustworthy AI: Efficient Algorithms for Building Probably Robust and Verifiable Neural Networks ".

## Footnotes

[1]For this result, the optimality assumes we abstain at points where the top-1 class has probability lower than 0.5.

[2] $D^{\alpha}$ is the multi-variate differential operator, where $\alpha$ is a multi-index.

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
