[Supplementary Material]

# Appendix

## A  Mathematical Preliminaries

In this section, we provide a brief introduction to the mathematical tools used in proving the theorems given in this paper :

**Log-Concave Functions [18]**   A non-negative function $f : \mathbb{R}^n \to \mathbb{R}^+$ is log-concave if it satisfies the inequality

$$f(\theta x + (1 - \theta)y) \geq f(x)^\theta f(y)^{1-\theta}$$

for all $x, y \in \mathbb{R}$ and $0 < \theta < 1$. Some popular examples of log-concave functions are the Gaussian pdf and $0 - 1$ indicator functions of convex sets. Log-concave functions satisfy the following properties:

- Log-concave functions are also quasi-concave.
- If $f, g$ are both log-concave functions then the convolution $f \star g$ is also log-concave.

**Quasi-Concave Functions [18]**   A function $f$ is called quasi-concave if

$$f(\lambda x + (1 - \lambda)y) \geq \min \big\{ f(x), f(y) \big\}$$

Quasi-concave functions satisfy the following properties:

- If $f$ is quasi-concave, then the superlevel sets, i.e., sets $S$ of the form $S = \{z \mid f(z) \geq \gamma\}$ for some $\gamma$, are convex.
- If $f_1, f_2, \ldots, f_n$ are both quasi-concave functions then the point-wise minimum of these, i.e., $f = \min_{1 \leq i \leq n} f_i$ is also quasi-concave.

**Absolute Moments Of Gaussian Random Variable**   : The $i^{th}$ absolute moments of a random variable $x$ is given as $\mathbb{E}[|x^i|]$. For Gaussian random variable with variance $\sigma^2$, the absolute moments are $\mathbb{E}[|x^i|] = \frac{\sigma^i 2^{i/2}}{\sqrt{\pi}} \Gamma(\frac{i+1}{2})$, where $\Gamma$ is the Gamma function. Some properties of the Gamma function are: $\Gamma(i) = (i-1)!$, $\Gamma(\frac{i+1}{2}) = \frac{i-1}{2}\Gamma(\frac{i-1}{2})$ for all $i \in \mathbb{N}_{>0}$.

**Subgaussian Random Varibale/Vector**   A random variable $x \in \mathbb{R}$ is said to be sub-Gaussian with parameter $\sigma^2$ if $\mathbb{E}[x] = 0$ and its moment generating function satisfies $\mathbb{E}[e^{sx}] \leq e^{\frac{\sigma^2 s^2}{2}}$. This is denoted as $x \sim \text{subG}(\sigma^2)$. Then a random vector $X$ is considered to be sub-Gaussian with parameter $\sigma^2$ if for all unit vectors $v$, $v^T X \sim \text{subG}(\sigma^2)$. With slight abuse of notation we denote this as $X \sim \text{subG}(\sigma^2)$.

Sub-Gaussian random variables satisfy the following properties:

- If $x_1, x_2, \ldots, x_N$ are independent sub-Gaussian random variables with parameter $k$, then $\frac{\sum_{i=1}^{N} x_i}{N} \sim \text{subG}(\frac{k}{N})$.
- If $x \sim \text{subG}(k)$, then $\mathbb{P}(x > t) \leq e^{-\frac{t^2}{2k}}$ and $\mathbb{P}(x < -t) \leq e^{-\frac{t^2}{2k}}$
- If $x_1, x_2, \ldots, x_N$ are not necessarily independent sub-Gaussian random variables with parameter $k$, then $\mathbb{P}(\max_{1 \leq i \leq N} x_i > t) \leq N e^{-\frac{t^2}{2k}}$.

**Generalized Neyman Pearson Lemma [17]**   In order to solve the optimization problems for our framework we use the Generalized Neymann Pearson Lemma [17]. Here, we give the lemma with a simplified short proof

**Lemma 1** (Generalized Neymann Pearson Lemma). *Let $f_0, f_1, \ldots, f_m$ be real-valued, $\mu-$integrable functions defined on a Euclidean space $X$ . Let $\psi_0$ be any function of the form*

$$\psi_0(x) \begin{cases} = 1, & \textit{if } f_0(x) < k_1 f_1(x) + \ldots + k_m f_m(x) \\ = \gamma(x), & \textit{if } f_0(x) = k_1 f_1(x) + \ldots + k_m f_m(x) \\ = 0, & \textit{if } f_0(x) > k_1 f_1(x) + \ldots + k_m f_m(x) \end{cases} \tag{8}$$

where $0 \leq \gamma(x) \leq 1$. *Then $\psi_0$ minimizes $\int_X \psi_0 f_0 d\mu$ over all $\psi, 0 \leq \psi \leq 1$ such that for $i = 1, \ldots, m$*

$$k_i \int_X \psi f_i d\mu \geq k_i \int_X \psi_0 f_i d\mu$$

*Proof.* We start by observing that under the given definition of $\psi_0, \psi$ the following inequality holds

$$\int_X (\psi - \psi_0)\left(f_0 - \sum_{i=1}^m k_i f_i\right) d\mu \geq 0 \tag{9}$$

We can show by proving $\forall x, (\psi(x) - \psi_0(x))\left(f_0(x) - \sum_{i=1}^m k_i f_i(x)\right) \geq 0$. We show this by doing a case analysis:

- If $f_0(x) - \sum_{i=1}^m k_i f_i(x)) > 0$ then $\psi_0(x) = 0$. As $\psi(x) \geq 0$, $\psi(x) - \psi_0(x) \geq 0$ making $(\psi(x) - \psi_0(x))\left(f_0(x) - \sum_{i=1}^m k_i f_i(x)\right) \geq 0$.

- If $f_0(x) - \sum_{i=1}^m k_i f_i(x)) < 0$ then $\psi_0(x) = 1$. As $\psi(x) \leq 1$, $\psi(x) - \psi_0(x) \leq 0$ making $(\psi(x) - \psi_0(x))\left(f_0(x) - \sum_{i=1}^m k_i f_i(x)\right) \geq 0$.

- Finally if $f_0(x) - \sum_{i=1}^m k_i f_i(x)) = 0$ then $(\psi(x) - \psi_0(x))\left(f_0(x) - \sum_{i=1}^m k_i f_i(x)\right) = 0 \geq 0$.

Using the inequality 9, we see that

$$\int_X \psi\left(f_0 - \sum_{i=1}^m k_i f_i\right) d\mu \geq \int_X \psi_0\left(f_0 - \sum_{i=1}^m k_i f_i\right) d\mu$$

$$\forall i \geq 1, \ k_i \int_X \psi f_i d\mu \geq k_i \int_X \psi_0 f_i d\mu \implies \int_X \psi f_0 d\mu \geq \int_X \psi_0 f_0 d\mu$$

$\square$

# B  Proofs for A General Framework for Randomized Smoothing

## B.1  Regularity Properties of $\mathcal{G}_{\mathcal{F}}^{\mu}$

**Theorem 1.** *If $\forall \alpha \in \mathbb{N}^d$, $\int_{\mathbb{R}^d} |D^\alpha \mu(z)| dz$ exists and is finite, then $\mathcal{G}_{\mathcal{F}}^{\mu} \subset \mathcal{C}^\infty$. Moreover if $g \in \mathcal{G}_{\mathcal{F}}^{\mu}$ is given as $g = f \star \mu$ for some $f \in \mathcal{F}$, then*

$$\nabla^i g(x) = \int_{\mathbb{R}^d} f(y)(-1)^i (\nabla^i \mu)(y - x) dy$$

*Proof.* It follows from the definition that whenever the right-hand side exists, we have

$$g(x) = \int_{\mathbb{R}^d} f(x + z)\mu(z) dz$$

$$= \int_{\mathbb{R}^d} f(y)\mu(y - x) dy,$$

$$\nabla_x^i g(x) = \nabla_x^i \int_{\mathbb{R}^d} f(y)\mu(y - x) dy$$

$$= \int_{\mathbb{R}^d} f(y)\nabla_x^i \mu(y - x) dy.$$

As we have $i > 1$, we get $\nabla_x^i (y - x) = 0$ and

$$\nabla_x^i g(x) = \int_{\mathbb{R}^d} f(y)(\nabla^i \mu)(y - x)(\nabla_x(y - x))^i dy$$

$$= \int_{\mathbb{R}^d} f(y)(-1)^i (\nabla^i \mu)(y - x) dy.$$

In order to show that the integral exists and is finite we show that integral converges for every element of the tensor. Our problem reduces to showing that for all $\alpha \in \mathbb{N}^d$ the integral $\int_{\mathbb{R}^d} f(y)(-1)^i (D^\alpha \mu)(y - x) dy$ exists and is finite. Using integrability conditions we see this is equivalent to showing

$$\int_{\mathbb{R}^d} |(D^\alpha \mu)(y - x)| dy < \infty \quad \Longleftrightarrow \quad \int_{\mathbb{R}^d} |D^\alpha \mu(z)| dz < \infty.$$

$\square$

**Lemma 2.** *Let $\mu_0(z)$ denote the Gaussian distribution $\mathcal{N}(0, \sigma^2)$, then $\frac{d^j}{dz^j} \mu_0(z) = q_j(z)\mu_0(z)$ for some $j^{th}$ degree polynomial $q_j$ with finite coefficient $a_{j,i}$ for $0 \le i \le j$. Moreover, $a_{j+1,i} = (i+1)a_{j,i+1} - \frac{1}{\sigma^2} a_{j,i-1}$*

*Proof.* For the base case $j = 1$, we see that $\frac{d}{dz}\mu_0(z) = -\frac{z}{\sigma^2}\mu_0(z)$. Thus, we we have $q_1(z) = -\frac{z}{\sigma^2}$ which is a degree 1 polynomial.

For the inductive step, we see that $\frac{d^{j+1}}{dz^{j+1}}\mu_0(z) = \frac{d}{dz}\left(\frac{d^j}{dz^j}\mu_0(z)\right) = \frac{d}{dz}q_j(z)\mu_0(z) = \left(\frac{d}{dz}q_j(z)\right)\mu_0(z) + q_j(z)\left(-\frac{z}{\sigma^2}\mu_0(z)\right) = \left(\frac{d}{dz}q_j(z) - \frac{z}{\sigma^2}q_j(z)\right)\mu_0(z)$. Thus, $q_{j+1}(z) = \left(\frac{d}{dz}q_j(z) - \frac{z}{\sigma^2}q_j(z)\right)$ which is clearly a polynomial of degree $(j+1)$ with coefficients $a_{j+1,i} = (i+1)a_{j,i+1} - \frac{1}{\sigma^2}a_{j,i-1}$. $\square$

**Corollary 1.1.** *When $\mu$ is given as the isotropic Gaussian distribution $\mathcal{N}(0, \sigma^2\mathcal{I})$, then $\mathcal{G}_{\mathcal{F}}^{\mu} \subset \mathcal{C}^\infty$ and $\nabla^i g(x) = \int_{\mathbb{R}^d} f(y)(-1)^i \nabla^i \mu(y - x) dy$.*

*Proof.* Using the fact that for isotropic gaussian we can write $\mu(x) = \Pi_{i=1}^d \mu_0(x_i)$ (where $\mu_0$ is also a gaussian pdf), we have

$$\int_{\mathbb{R}^d} |D^\alpha \mu(z)| dz = \int_{\mathbb{R}} \left| D^\alpha \Pi_{i=1}^d \mu_0(z_i) \right| dz$$

$$= \int_{\mathbb{R}} \Pi_{i=1}^d \left| \frac{d^{\alpha_i}}{dz_i^{\alpha_i}} \mu_0(z_i) \right| dz$$

$$= \Pi_{i=1}^d \int_{\mathbb{R}} \left| \frac{d^{\alpha_i}}{dz_i^{\alpha_i}} \mu_0(z_i) \right| dz_i.$$

As we know that the product of $d$ finite values is finite, we only need to show that for any value $i \in \mathbb{N}$, $\int_{\mathbb{R}} |\frac{d^i}{dz^i} \mu_0(z)| dz$ is finite. Using Lemma 2, we see that it suffices to show that for all $j \in \mathbb{N}$, $\int_{\mathbb{R}} |x^j \mu_0(z)| dz$ is finite. But this quantity is known as the absolute central moment of normal distribution and is given by $\sigma^j \sqrt{\frac{2^j}{\pi}} \Gamma(\frac{1+j}{2})$ which is finite. $\qquad \square$

**Lemma 3.** *For $\mu$ given by isotropic Gaussian distribution $\mathcal{N}(0, \sigma^2 \mathbf{I})$ and a function $f \in \mathcal{F}$, if $\|x - w\|_\infty \leq R$ for some finite $R$, then*

$$\int_{\mathbb{R}^d} f(y) \sum_{\alpha \in \mathbb{N}^d} D^\alpha \mu(y - w) \frac{(w - x)^\alpha}{\alpha!} dy = \sum_{\alpha \in \mathbb{N}^d} \frac{(w - x)^\alpha}{\alpha!} \int_{\mathbb{R}^d} f(y) D^\alpha \mu(y - w) dy$$

*Proof.* As this can be regarded as a double integral where the sum is an integral over the counting measure, we can use Fubini's Theorem to reduce it to proving :

$$\sum_{\alpha \in \mathbb{N}^d} \left| \frac{(w - x)^\alpha}{\alpha!} \right| \left| \int_{\mathbb{R}^d} |f(y) D^\alpha \mu(y - w) dy| \right| < \infty$$

As $f(y)$ only takes values between 0 and 1, we see that

$$\sum_{\alpha \in \mathbb{N}^d} \left| \frac{(w - x)^\alpha}{\alpha!} \right| \left| \int_{\mathbb{R}^d} |f(y) D^\alpha \mu(y - w)| dy \right| \leq \sum_{\alpha \in \mathbb{N}^d} \left| \frac{(w - x)^\alpha}{\alpha!} \right| \left| \int_{\mathbb{R}^d} |D^\alpha \mu(z)| dz \right|$$

$$= \sum_{\alpha \in \mathbb{N}^d} \prod_{j=1}^d \left( \left| \frac{(w_j - x_j)^{\alpha_j}}{\alpha_j!} \right| \left| \int_{\mathbb{R}} \left| \frac{d^{\alpha_j}}{dz^{\alpha_j}} \mu_0(z) \right| dz \right| \right)$$

$$= \prod_{j=1}^d \left( \sum_{k \in \mathbb{N}} \left| \frac{(w_j - x_j)^k}{k!} \right| \left| \int_{\mathbb{R}} \left| \frac{d^k}{dz^k} \mu_0(z) \right| dz \right| \right)$$

$$\leq \left( \sum_{k \in \mathbb{N}} \frac{R^k}{k!} \int_{\mathbb{R}} \left| \frac{d^k}{dz^k} \mu_0(z) \right| dz \right)^d$$

As $d$ is finite number, it is sufficient to show that the infinite sum converges. Using Lemma 2, we see that $\frac{d^k}{dz^k} \mu_0(z) = q_k(z) \mu_0(z)$ for some $k^{th}$ degree polynomial $q_k$. Let $a_{j,i}$ be the co-efficient of $z^i$ in the polynomial $q_j$.

$$\int_{\mathbb{R}} \left| \frac{d^k}{dz^k} \mu_0(z) \right| dz = \int_{\mathbb{R}} |q_k(z) \mu_0(z)| dz \leq \sum_{i=0}^k \int_{\mathbb{R}} |a_{ki} z^i \mu_0(z)| dz = \sum_{i=0}^k |a_{ki}| \frac{\sigma^i 2^{i/2} \Gamma(\frac{i+1}{2})}{\sqrt{\pi}}$$

Using comparison condition, it is sufficient to show that the sum $\sum_{k=0}^\infty \frac{R^k}{k!} \sum_{i=0}^k |a_{k,i}| \frac{\sigma^i 2^{i/2} \Gamma(\frac{i+1}{2})}{\sqrt{\pi}}$ converges. Now, we prove the convergence using the ratio test. Using the fact $a_{j+1,i} = (i + 1) a_{j,i+1} - \frac{1}{\sigma^2} a_{j,i-1}$, we see

$$\frac{\frac{R^{k+1}}{(k+1)!} \sum_{i=0}^{k+1} |a_{k+1,i}| \frac{2^{i/2} \sigma^{i+1} \Gamma(\frac{i+1}{2})}{\sqrt{\pi}}}{\frac{R^k}{k!} \sum_{i=0}^k |a_{k,i}| \frac{2^{i/2} \sigma^{i+1} \Gamma(\frac{i+1}{2})}{\sqrt{\pi}}} = \frac{R}{k+1} \frac{\sum_{i=0}^{k+1} |a_{k+1,i}| 2^{i/2} \sigma^{i+1} \Gamma(\frac{i+1}{2})}{\sum_{i=0}^k |a_{k,i}| 2^{i/2} \sigma^{i+1} \Gamma(\frac{i+1}{2})}$$

$$\leq \frac{R}{k+1} \frac{\sum_{i=0}^{k+1} ((i+1)|a_{k,i+1}| + \frac{1}{\sigma^2}|a_{k,i-1}|) 2^{i/2} \sigma^{i+1} \Gamma(\frac{i+1}{2})}{\sum_{i=0}^k |a_{k,i}| 2^{i/2} \sigma^{i+1} \Gamma(\frac{i+1}{2})}$$

$$= \frac{R}{k+1} \frac{\sum_{i=0}^k |a_{k,i}| (i \sigma^i \Gamma(\frac{i}{2}) 2^{(i-1)/2} + \sigma^i \Gamma(\frac{i+2}{2}) 2^{(i+1)/2})}{\sum_{i=0}^k |a_{k,i}| 2^{i/2} \sigma^{i+1} \Gamma(\frac{i+1}{2})}$$

$$= \frac{R}{k+1} \frac{\sum_{i=0}^{k} 2i|a_{k,i}|2^{(i-1)/2}\sigma^i\Gamma(\frac{i}{2})}{\sum_{i=0}^{k} |a_{k,i}|2^{i/2}\sigma^{i+1}\Gamma(\frac{i+1}{2})}$$

Using ([19]), $\frac{\Gamma(\frac{i}{2})}{\Gamma(\frac{i+1}{2})} \leq \frac{\sqrt{\frac{i+1}{2}}}{\frac{i}{2}}$,

$$\leq \frac{R}{k+1} \frac{\sum_{i=0}^{k} 2\sqrt{i+1}|a_{k,i}|2^{i/2}\sigma^i\Gamma(\frac{i+1}{2})}{\sum_{i=0}^{k} \sigma|a_{k,i}|2^{i/2}\sigma^i\Gamma(\frac{i+1}{2})}$$

$$\leq \frac{R}{k+1} \frac{2\sqrt{k+1}}{\sigma} = \frac{2R}{\sigma\sqrt{k+1}}$$

For any finite $R$, $\lim_{k\to\infty} \frac{2R}{\sigma\sqrt{k+1}} = 0 < 1$. So the series is convergent. $\qquad\square$

**Theorem 2.** *When $\mu$ is the isotropic Gaussian distribution $\mathcal{N}(0,\sigma^2\mathcal{I})$, then $\forall g \in \mathcal{G}_{\mathcal{F}}^\mu$, $g$ is a real analytic function with infinite radius of convergence ,i.e., the Taylor series of $g$ around any point $w$ converges to the function $g$ everywhere.*

*Proof.* Let us take the Taylor expansion at a point $w$. In order to show that the Taylor expansion has an infinite radius of convergence, we consider any arbitrarily big value $R$ and show that if $\|x - w\|_\infty \leq R$, then

$$g(x) = \int_{\mathbb{R}^d} f(x+z)\mu(z)dz$$

$$= \int_{\mathbb{R}^d} f(y)\mu(y-x)dy$$

Using the Taylor expansion of the gaussian PDF and the fact it's radius of convergence is infinite

$$g(x) = \int_{\mathbb{R}^d} f(y) \sum_{\alpha\in\mathbb{N}^d} D^\alpha\mu(y-w)\frac{((y-x)-(y-w))^\alpha}{\alpha!}dy$$

Now using Lemma 3, we get

$$g(x) = \sum_{\alpha\in\mathbb{N}^d} \frac{(w-x)^\alpha}{\alpha!} \int_{\mathbb{R}^d} f(y)D^\alpha\mu(y-w)dy$$

Finally, we use Corollary 1.1, to get

$$g(x) = \sum_{\alpha\in\mathbb{N}^d} \frac{(x-w)^\alpha}{\alpha!} D^\alpha g(w)$$

As for any arbitrarily large $R$, the Taylor series converges for any $x$ satisfying $\|x - w\|_\infty \leq R$ we see that the radius of convergence is infinite. Clearly, this holds for all points $w \in \mathbb{R}^d$ and all $g \in \mathcal{G}_{\mathcal{F}}^\mu$. $\quad\square$

## B.2 Certification For Randomized Smoothing Using First-Order Information

We use the Generalized Neymann Pearson Lemma ([17]) to solve the optimization problem 2.

**Theorem 3** (Lower Bound of $\mathbf{p}_x(z)$)**.** *For a base classifier $f \in \mathcal{F}$, if $g = f \star \mu$, $\mu$ is the isotropic Gaussian distribution $\mathcal{N}(0,\sigma^2\mathcal{I})$, $y^{(0)} = g(x)$, $y^{(1)} = \nabla g(x)$, then for any unit vector $v$ and any positive value of $r$, $\mathbf{p}_x(x + \sigma r v)$ can be lower bounded by solving the following set of equations:*

$$\int_{-\infty}^{\infty} \frac{1}{\sqrt{2\pi}}e^{-\frac{x^2}{2}}\Phi(c(x))dx = q \tag{6}$$

$$\int_{-\infty}^{\infty} \frac{1}{\sqrt{2\pi}}e^{-\frac{x^2}{2}}\frac{1}{\sqrt{2\pi}}e^{-\frac{c(x)^2}{2}}dx = m_2 \quad \text{(7a)} \qquad \int_{-\infty}^{\infty} \frac{1}{\sqrt{2\pi}}xe^{-\frac{x^2}{2}}\Phi(c(x))dx = m_1 \quad \text{(7b)}$$

*with $q \leq y^{(0)}, m_1 \leq \sigma v^T y^{(1)}, m_2 \leq \sigma\|y^{(1)} - v^T y^{(1)}v\|_2$, $c(x) := c_0 + c_1 x + c_2 e^{rx}$, and $\Phi(z)$ being the CDF of the standard normal distribution. If the solution $(c_0, c_1, c_2)$ of above equations has $c_1 < 0$, then the lower bound of $\mathbf{p}_x(x + \sigma r v)$ is instead given by solving Equations (4) to (5a) with $c(x) := c_0 + c_2 e^{rx}$.*

*Proof.* In order to solve the Equation 2 under the local constraints $H_1^{x_0}(h) = y^{(0)} - h(x_0) = 0$ and $H_2^{x_0}(h) = y^{(1)} - \nabla h(x_0) = 0$. Setting the measure to be $\mu_\sigma = \left(\frac{1}{\sqrt{2\pi\sigma^2}}\right)^{\frac{d}{2}} e^{-\frac{\|z\|_2^2}{2\sigma^2}}$ and using the fact that $h \in \mathcal{G}_{\mathcal{F}}^\mu$, we have $h = f' \star \mu$ for some $f \in \mathcal{F}$. Thus, the constraints can be expressed using the base classifier as given as

$$\int_{\mathbb{R}^d} f'(x_0 + z) d\mu_\sigma = y^{(0)}$$

$$\int_{\mathbb{R}^d} \frac{z}{\sigma^2} f'(x_0 + z) d\mu_\sigma = y^{(1)}$$

and the optimization problem is given as $\min_{f' \in \mathcal{F}} \int_{\mathbb{R}^d} e^{-\frac{R^2}{2\sigma^2}} e^{\frac{R v^T z}{\sigma^2}} f'(x_0 + z) d\mu_\sigma$.

In order to make the math simpler we use the following basis transformation we rotate the basis such that we have $z_1$ along the $v$, $z_2$ along $y^{(1)} - v^T y^{(1)} v$ and then we scale the basis by a factor of $\frac{1}{\sigma}$.

The constraints can now be expressed using $\mu = \left(\frac{1}{\sqrt{2\pi}}\right)^{\frac{d}{2}} e^{-\frac{\|z\|_2^2}{2}}$ given as

$$\int_{\mathbb{R}^d} f'(x_0 + z) d\mu = y^{(0)}$$

$$\int_{\mathbb{R}^d} \frac{z_1}{\sigma} f'(x_0 + z) d\mu = v^T y^{(1)}$$

$$\int_{\mathbb{R}^d} \frac{z_2}{\sigma} f'(x_0 + z) d\mu = \left\| y^{(1)} - v^T y^{(1)} v \right\|_2$$

$$\int_{\mathbb{R}^d} \frac{z_i}{\sigma} f'(x_0 + z) d\mu = 0, \text{ if } i \geq 3$$

Then defining $r = \frac{R}{\sigma}$, the optimization problem is given as

$$\min_{f' \in \mathcal{F}} \int_{\mathbb{R}^d} e^{-\frac{r^2}{2}} e^{r z_1} f'(x_0 + z) d\mu = e^{-\frac{r^2}{2}} \min_{f' \in \mathcal{F}} \int_{\mathbb{R}^d} e^{r z_1} f'(x_0 + z) d\mu$$

Using the Generalized Neymann Pearson Lemma, we see that the minima occurs for the function $f_0$ such that $f_0(x_0 + z) = 1$ if $e^{r z_1} \leq a^T z + b$ and 0 otherwise, for some $a \in \mathbb{R}^d, b \in \mathbb{R}$ such that the constraints are satisfied. We can use the constraints to solve for $a, b$ in order to get the value of the minimization.

**Claim.** *For $i \geq 3$, we can show that we need $a_i = 0$.*

*Proof.* Assume to the contrary $a_i > 0$, then

$$\int_{\mathbb{R}^d} \frac{z_i}{\sigma} f'(x_0 + z) d\mu = \int_{e^{r z_1} \leq a^T z + b} \frac{z_i}{\sigma} f'(x_0 + z) d\mu$$

$$= \int_{z_i \geq \frac{e^{r z_1} - (a_{-i}^T z_{-i} + b)}{a_i}} \frac{z_i}{\sigma} d\mu > 0$$

Consider the function $l(z_{-i}) := \frac{e^{r z_1} - (a_{-i}^T z_{-i} + b)}{a_i}$. We know that for the standard normal measure $\mu_0$ gaussian cdf, the value of $\int_{z_i \geq l} \frac{z_i}{\sigma} d\mu_0 \geq 0$ for any value of $l$ with the equality holding only if $l = -\infty$. So, we see that $\int_{z_i \geq l(z_{-i})} \frac{z_i}{\sigma} d\mu \geq 0$ for any function $l(z_{-i})$ with equality holding only if $l(z_{-i})$ is $-\infty$ almost everywhere does not hold. So, we get a contradiction. Similarly we can also show a contradiction for the case when $a_i < 0$. Thus, we have that for $i \geq 3$, $a_i = 0$. $\square$

Substituting the values of $a_i$ in our constraints and simplifying the integrals we have the following system of equations:

$$p_{x_0}(x_0 + \sigma r v) = \int_{-\infty}^{\infty} \frac{1}{\sqrt{2\pi}} e^{-\frac{(x-r)^2}{2}} \Phi(c(x)) dx$$

$$\int_{-\infty}^{\infty} \frac{1}{\sqrt{2\pi}} e^{-\frac{x^2}{2}} \Phi(c(x)) dx = y^{(0)}$$

$$\int_{-\infty}^{\infty} \frac{1}{\sqrt{2\pi}} e^{-\frac{x^2}{2}} \frac{1}{\sqrt{2\pi}} e^{-\frac{c(x)^2}{2}} dx = \sigma \left\| y^{(1)} - v^T y^{(1)} v \right\|_2$$

$$\int_{-\infty}^{\infty} \frac{1}{\sqrt{2\pi}} x e^{-\frac{x^2}{2}} \Phi(c(x)) dx = \sigma v^T y^{(1)}$$

where $c(x) := \frac{b}{a_2} + \frac{a_1}{a_2} x + \frac{-1}{a_2} e^{rx}$ and $\Phi(z)$ denotes the CDF of the standard normal distribution.

Although this gives a solution for $p_{x_0}(x_0 + \sigma r v)$ the constraints here have equalities which require us to get exact values of $y^{(0)}, y^{(1)}$. This is not possible to achieve in practice. In practice, we can only get a high confidence interval estimate of the values. So, we need to be able to solve for $p_{x_0}(x_0 + \sigma r v)$ given interval estimates of the parameters.

We notice that using the same argument in the Claim, we can use the constraint $\int_{-\infty}^{\infty} \frac{1}{\sqrt{2\pi}} e^{-\frac{x^2}{2}} \frac{1}{\sqrt{2\pi}} e^{-\frac{c(x)^2}{2}} dx \geq 0$ to show that $a_2 > 0$. Similarly we have that if $y^{(0)} > 0$, then $b > 0$. Otherwise if $b < 0$, then we see that $c(x) < \frac{a_1}{a_2} x$ giving $\int_{-\infty}^{\infty} \frac{1}{\sqrt{2\pi}} e^{-\frac{x^2}{2}} \Phi(c(x)) dx < \int_{-\infty}^{\infty} \frac{1}{\sqrt{2\pi}} e^{-\frac{x^2}{2}} \Phi(\frac{a_1}{a_2} x) dx = 0.5$. As the coefficients $a_2, b > 0$, Generalized Neymann Pearson Lemma allows us to use lower bounds $p \leq y^{(0)}$ and $m_2 \leq \sigma \left\| y^{(1)} - v^T y^{(1)} v \right\|_2$ in the constraints to still get a valid estimate of $p_{x_0}(x_0 + \sigma r v)$.

The only variable that can be both negative and positive is $a_1$. If we use a lower bound $m_1 \leq \sigma v^T y^{(1)}$ and the resulting solution has a positive value of $a_1$ then it is valid. However, we see that if we get a negative value of $a_1$ in the solution we can instead solve the relaxed minimization problem to get a lower bound of $p_{x_0}(x_0 + \sigma r v)$ without the constraint 5b. We give this as

$$p_{x_0}(x_0 + \sigma r v) = \int_{-\infty}^{\infty} \frac{1}{\sqrt{2\pi}} e^{-\frac{(x-r)^2}{2}} \Phi(c(x)) dx$$

$$\int_{-\infty}^{\infty} \frac{1}{\sqrt{2\pi}} e^{-\frac{x^2}{2}} \Phi(c(x)) dx = y^{(0)}$$

$$\int_{-\infty}^{\infty} \frac{1}{\sqrt{2\pi}} e^{-\frac{x^2}{2}} \frac{1}{\sqrt{2\pi}} e^{-\frac{c(x)^2}{2}} dx = \sigma \left\| y^{(1)} - v^T y^{(1)} v \right\|_2$$

where $c(x) := \frac{b}{a_2} + \frac{-1}{a_2} e^{rx}$.  $\square$

**Proposition 1.** *The certified safety region, $\mathbf{SR}_L(x)$, calculated using the zeroth and first-order local information is convex, i.e., if $x_1, x_2 \in \mathbf{SR}_L(x)$ then $\frac{x_1 + x_2}{2} \in \mathbf{SR}_L(x)$.*

*Proof.*

$$\mathbf{SR}_L(x) = \{z \mid \mathbf{p}_x(z) > 0.5\}$$

So, $\mathbf{SR}_L(x)$ is a superlevel set of $\mathbf{p}_x$. In order to show $\mathbf{SR}_L(x)$ is convex it is sufficient to show $\mathbf{p}_x$ is a quasi-concave function. Using the definition of $\mathbf{p}_x$, we have

$$\mathbf{p}_x(z) = \min_{h \in \mathcal{G}_{\mathcal{F}}^{\mu}} h(z)_c \quad \textbf{s.t.} \quad h(x) = y^{(0)}, \nabla h(x) = y^{(1)}$$

**Claim.** *For the lower bound probability function $\mathbf{p}_x$ calculated using the zeroth and first-order information, $\mathbf{p}_x(z) = (f \star \mu)(z)$ for some $f$ such that $f \star \mu$ satisfies all the optimization constraints and $f \star \mu$ is quasi-concave.*

*Proof.* Using Generalized Neyman Pearson Lemma, we see that the minima of the constrained optimization problem occurs for some function $f$ that satisfies $f(x) = 1$ if $e^{\frac{2z^T x - \|z\|_2^2}{2\sigma^2}} \leq a^T x + b$ and 0 otherwise. Thus, $\mathbf{p}_x(z) = (f \star \mu)(z)$ where $f$ is the indicator function for the set

$$S = \{x \mid x \in \mathbb{R}^d; e^{\frac{2z^T x - \|z\|_2^2}{2\sigma^2}} \leq a^T x + b\}$$

It is easy to see that the function $e^{\frac{2z^T x - \|z\|_2^2}{2\sigma^2}} - a^T x - b$ is convex as the Hessian is given as $\frac{zz^T + \sigma^2 I}{\sigma^4} e^{\frac{2z^T x - \|z\|_2^2}{2\sigma^2}} \succ 0$. Thus, the set $S$ being a level set of a convex function is also convex. So, $f$ is the indicator function of a convex set and thus a log-concave function. Moreover, we have that $\mu$ being isotropic gaussian distribution is also log-concave. From the properties of log-concave functions we get that the convolution $f \star \mu$ is also log-concave and as log-concave functions are also quasi-concave, $f \star \mu$ is quasi-concave. $\qquad\square$

Using this claim we see that as at every point $z$, $\mathbf{p}_x(z) = (f \star \mu)(z)$ for some $f \star \mu$ that satisfies all the constraints and is also quasi-concave, we can add an extra constraint to get

$$\mathbf{p}_x(z) = \min_{g \in \mathcal{G}_{\mathcal{F}}^\mu} g(z)_c \quad \textbf{s.t.} \quad g(x) = y^{(0)}, \nabla g(x) = y^{(1)}, \quad g(x)_c \textbf{ is quasi-concave}$$

to get the same $\mathbf{p}_x$. As $\mathbf{p}_x$ can be written as the minima over a set of quasi-concave functions, we see that by the property of quasi-concave functions, $\mathbf{p}_x$ is also quasi-concave. Thus, we see that $\mathbf{SR}_L(x)$ is convex. $\qquad\square$

**Proposition 2.** *For any given value of $y^{(0)}, y^{(1)}$, the directional robustness along $v$, $\mathbf{R}_v$, given by the first-order certification method is a non-increasing function of the angle between $v$ and $y^{(1)}$, i.e., $\cos^{-1}\left(\frac{v^T y^{(1)}}{\|v\|_2 \|y^{(1)}\|_2}\right)$.*

*Proof.* It follows from Theorem 3 that given some value of $y^{(0)}$ and $y^{(1)}$ the minimal probability $p_{x_0}(x_0 + rv)$ at distance $r$ along a direction $v$ depends only on the angle between $v$ and $y^{(1)}$. Given some fixed $y^{(0)}, y^{(1)}$ we can write $p_{x_0}(x_0 + rv)$ as a function of $\theta$, $p_{x_0}(x_0 + rv) = p_x(r, \theta)$. Given this we claim

**Claim.** *For any given value of $r$, $\frac{\partial p(r,\theta)}{\partial \theta} \leq 0$.*

*Proof.* As it is easier to state our theorems for vectors, We relate $\theta$ back to our vectors using a vector valued function $w(\alpha)$ that gives us vectors in some plane $P$ containing $y^{(1)}$ such that the angle between $y^{(1)}$ and $w(\alpha)$ is $\alpha$. Now given any angle $\alpha$ and some distance $r$, Theorem 3 gives us that there exists some function $f_0$ such that for $g_0 = f_0 \star \mu$ all the local constraints are satisfied and $g_0(x_0 + rw(\alpha)) = p(r, \alpha)$. As $p(r, \theta)$ gives the minimum value that can be assigned $x_0 + rw(\theta)$ by a function satisfying the given constraints, we see the $p(r, \theta) \leq g_0(x_0 + rw(\theta))$. So we see that if $\frac{\partial g_0(x_0 + rw(\theta))}{\partial \theta}\big|_{\theta=\alpha} \leq 0$ then $\frac{\partial p(r,\theta)}{\partial \theta}\big|_{\theta=\alpha} \leq 0$. It is sufficient to show that $\frac{\partial g_0(x_0 + rw(\theta))}{\partial \theta}\big|_{\theta=\alpha} \leq 0$. In order to make the make the calculation simpler we can do the same basis transformation as in proof of Theorem 3. Under the new basis we have $\frac{\partial g_0(x_0 + rw(\theta))}{\partial \theta}\big|_{\theta=\alpha} = -\frac{\partial g_0(x_0 + (z_1, z_2))}{\partial z_2}\big|_{z=(r,0)}$ where $(z_1, z_2)$ is a two dimensional vector $z_1$ along the old $w(0)$ and $z_2$ along $w(\frac{\pi}{2})$.

For the new basis we see that the proof of Theorem 3 also gives the form of $f_0$, i.e., there exist some constants $a_1, b \in \mathbb{R}$ and $a_2 \in \mathbb{R}^+$ such that $f_0(x_0 + z) = 1$ if $e^{rz_1} \leq a_1 z_1 + a_2 z_2 + b$ and 0 otherwise. Using this form we have

$$\frac{\partial g_0(x_0 + (z_1, z_2))}{\partial z_2}\bigg|_{z=(r,0)} = \frac{\partial}{\partial z_2} \int\int_{e^{rx} \leq a_1 x + a_2 y + b} \frac{1}{2\pi} e^{-\frac{(x-z_1)^2 + (y-z_2)^2}{2}} dy\, dx \bigg|_{z=(r,0)}$$

$$= \int\int_{e^{rx} \leq a_1 x + a_2 y + b} \frac{-y}{2\pi} e^{-\frac{(x-r)^2 + (y)^2}{2}} dy\, dx$$

$$= \int_{-\infty}^{\infty} \int_{\frac{e^{rx} - a_1 x - b}{a_2} \leq y} \frac{-y}{2\pi} e^{-\frac{(x-r)^2 + (y)^2}{2}} dy\, dx$$

$$= \int_{-\infty}^{\infty} \frac{1}{2\pi} e^{-\frac{(x-r)^2}{2}} - e^{-\frac{(e^{rx} a_1 x - b)^2}{2a_2^2}} dx \geq 0$$

$$\frac{\partial g_0(x_0 + rw(\theta))}{\partial \theta}\bigg|_{\theta=\alpha} = -\frac{\partial g_0(x_0 + (z_1, z_2))}{\partial z_2}\bigg|_{z=(r,0)} \leq 0$$

$$\implies \frac{\partial p(r,\theta)}{\partial \theta}\bigg|_{\theta=\alpha} \leq 0$$

$\square$

Using the claim we can show that $\mathbf{R_v}$ is a non-increasing function of the angle $\cos^{-1}\left(\frac{v^T y^{(1)}}{\|v\|_2\|y^{(1)}\|_2}\right)$ as follows: For angle $\alpha$ let $\mathbf{R_{w(\alpha)}} = r$, then $p_{x_0}(x_0 + rw(\alpha)) = 0.5$. Using the claim we see that for any value of $\beta > \alpha$, $p_{x_0}(x_0 + rw(\beta)) = p(r, \beta) \leq p(r, \alpha) = 0.5$. So we conclude that for any $\beta > \alpha$, $\mathbf{R_{w(\beta)}} \leq r = \mathbf{R_{w(\alpha)}}$. $\square$

**Corollary 3.1** (Certified $\ell_2$ Norm Radius). *For a base classifier $f \in \mathcal{F}$, if $g = f \star \mu$, where $\mu$ is the isotropic Gaussian distribution $\mathcal{N}(0, \sigma^2 \mathcal{I})$, $y^{(0)} = g(x)$, $y^{(1)} = \nabla g(x)$, the $\ell_2$ norm radius $R$ is given as $R = \sigma r$, where $(r, z_1, z_2)$ is the solution of the system of equations:*

$$\Phi(z_1 - r) - \Phi(z_2 - r) = 0.5 \tag{8}$$

$$\Phi(z_1) - \Phi(z_2) = q \tag{9a} \qquad \frac{1}{\sqrt{2\pi}}e^{-\frac{z_2^2}{2}} - \frac{1}{\sqrt{2\pi}}e^{-\frac{z_1^2}{2}} = m_1 \tag{9b}$$

*with $q \leq y^{(0)}$ and $m_1 \geq \sigma\|y^{(1)}\|_2$.*

*Proof.* Using Proposition 2 we see that the minimum value of $\mathbf{R_v}$ occurs when $v^T y^{(1)}$ is smallest. As $v^T y^{(1)} \geq -\|v\|_2\|y^{(1)}\|_2$ with equality when $v = \frac{-y^{(1)}}{\|y^{(1)}\|_2}$. Using Theorem 3 to solve for $\mathbf{R_v}$ along this direction yields $m_2 = 0$. Using the same proof as in the first Claim in proof of Theorem 3 we have $a_2 = 0$. So we can rewrite $f_0(x) = 1$ if $e^{rx_1} \leq a_1 x_1 + b$. Solving for $a_1, b$ under the constraints gives us the equations:

$$\int_{e^{rx_1} \leq a_1 x_1 + b} \frac{1}{\sqrt{2\pi}}e^{-\frac{x^2}{2}} dx = p$$

$$\int_{e^{rx_1} \leq a_1 x_1 + b} \frac{1}{\sqrt{2\pi}}xe^{-\frac{x^2}{2}} dx = -m_1$$

where As for all values of $a, b$ the solution to the equation $e^{rx_1} \leq a_1 x_1 + b$ is an interval of the form $[z_2, z_1]$, we can re-write the constraints as

$$\Phi(z_1) - \Phi(z_2) = p$$

$$\frac{1}{\sqrt{2\pi}}(e^{-\frac{z_2^2}{2}} - e^{-\frac{z_1^2}{2}}) = m_1$$

Using the resulting $f_0$ the minimum value of $g_0$ at $r$ is given as $\Phi(z_1 - r) - \Phi(z_2 - r)$ which can be equated to $0.5$ to give the radius. $\square$

**Corollary 3.2** (Certified $\ell_1$ Norm Radius). *For a base classifier $f \in \mathcal{F}$, if $g = f \star \mu$, where $\mu$ is the isotropic Gaussian distribution $\mathcal{N}(0, \sigma^2 \mathcal{I})$, $y^{(0)} = g(x)$, $y^{(1)} = \nabla g(x)$, the $\ell_1$ norm radius $R$ is obtained by solving $\mathbf{p}_x(x + Rv) = 0.5$, where $\mathbf{p}_x(x + Rv)$ is given by solving the problem in Theorem 3 with $m_1 \leq -\sigma\|y^{(1)}\|_\infty$, $m_2 \leq \sigma\sqrt{\|y^{(1)}\|_2^2 - \|y^{(1)}\|_\infty^2}$.*

*Proof.* We see that if the minimum directional robustness among the basis vectors is given as $R_{\min} = \min_i \left(\min(\mathbf{R}_{e_i}, \mathbf{R}_{-e_i})\right)$, then along every basis vector direction $\mathbf{R_v} \geq R_{\min}$. Thus, the points $\{R_{\min}e_i, -R_{\min}e_i \mid 1 \leq i \leq d\} \subset \mathbf{SR}_L$ and by Proposition 1, the convex hull of these points the $\ell_1$ norm ball of radius $R_{\min}$ is also in $\mathbf{SR}_L$. Thus, the $\ell_1$ norm certified radius can be given as $\min_i \left(\min(\mathbf{R}_{e_i}, \mathbf{R}_{-e_i})\right)$.

Using Proposition 2, we see that this minimum occurs in the direction with the largest angle with $y^{(1)}$. So, the projection of $y^{(1)}$ along this direction can be given as $\min_i \min(e_i^T y^{(1)}, -e_i^T y^{(1)}) = -\max_i \max(-e_i^T y^{(1)}, e_i^T y^{(1)}) = -\|y^{(1)}\|_\infty$. Now, we can use Theorem 3 to give us the final solution. $\square$

**Corollary 3.3** (Certified $\ell_\infty$ Norm Radius). *For a base classifier $f \in \mathcal{F}$, if $g = f \star \mu$, where $\mu$ is the isotropic Gaussian distribution $\mathcal{N}(0, \sigma^2 \mathcal{I})$, $y^{(0)} = g(x)$, $y^{(1)} = \nabla g(x)$, the $\ell_\infty$ norm radius $R$ is obtained by solving $\mathbf{p}_x(x + Rv) = 0.5$, where $\mathbf{p}_x(x + Rv)$ is given by solving the problem in Theorem 3 with $m_1 \leq -\frac{\sigma}{\sqrt{d}}\|y^{(1)}\|_1$, $m_2 \leq \frac{\sigma}{\sqrt{d}}\sqrt{d\|y^{(1)}\|_2^2 - \|y^{(1)}\|_1^2}$.*

*Proof.* Consider the set of vectors $S = \{v \mid |v_i| = \frac{1}{\sqrt{d}}\}$. We see that if the minimum directional robustness among the vectors in $S$ is given as $R_{\min} = \min_{v \in S} \mathbf{R}_v$, then along every vector direction $v$ in $S$, $\mathbf{R}_v \geq R_{\min}$. Thus, the points $\{R_{\min}v \mid v \in S\} \subset \mathbf{SR}_L$ and by Proposition 1, the convex hull of these points the $\ell_\infty$ norm ball of radius $R_{\min}$ is also in $\mathbf{SR}_L$. Thus, the $\ell_\infty$ norm certified radius can be given as $\min_{v \in S} \mathbf{R}_v$.

Using Proposition 2, we see that this minimum occurs in the direction with the largest angle with $y^{(1)}$. So, the projection of $y^{(1)}$ along this direction can be given as $\min_{v \in S} v^T y^{(1)} = -\max_{v \in S}(-v)^T y^{(1)} = -\max_{v \in S} v^T y^{(1)} = -\frac{1}{\sqrt{d}}\|y^{(1)}\|_1$. Now, we can use Theorem 3 to give us the final solution. $\qquad\square$

**Corollary 3.4** (Subspace Certified $\ell_p$ norm radius). *For a base classifier $f \in \mathcal{F}$, if $g = f \star \mu$, where $\mu$ is the isotropic Gaussian distribution $\mathcal{N}(0, \sigma^2 \mathcal{I})$, $y^{(0)} = g(x)$, $y^{(1)} = \nabla g(x)$, and a subspace $S$ with orthogonal projection matrix $P_S$, for $p = 1, 2, \infty$ the subspace $\ell_p$ norm certified radius $R$ is obtained by solving $\boldsymbol{p}_x(x + Rv) = 0.5$, where $\boldsymbol{p}_x(x + Rv)$ is given by solving the problem in Theorem 3 with $m_1 \leq -\sigma\|P_S y^{(1)}\|_{p'}, m_2 \leq \sigma\sqrt{\|y^{(1)}\|_2^2 - \|P_S y^{(1)}\|_{p'}^2}$, and $\|\cdot\|_{p'}$ is the dual norm of $\|\cdot\|_p$.*

*Proof.* For orthogonal projection $P_S$ onto a subspace $S$, we can consider the vector $P_S y^{(1)}$ instead of $y^{(1)}$, the using almost identical arguments as before we get the corresponding projections and we can solve for the certified radii using Theorem 3. $\qquad\square$

# C   Theoretical Case Study: Binary Linear Classifier

Figure 3: Certified safety regions for binary linear classifiers (input point $x$ grey circle at origin)

Given any binary linear classifier, $f(x) = \mathbf{1}_{w^T x + b \leq 0}$ let $g$ defined by $f \star \mu$ where $\mu$ is the isotropic Gaussian distribution $\mathcal{N}(0, \sigma^2 \mathbf{I})$ (for any $\sigma$) be the smoothed classifier. For this case, Cohen et al. [5, Appendix B] showed the following:

- The prediction of $g$ is same as the prediction of $f$, i.e., $\forall\, x \in \mathbb{R}^d$, $f(x) = 1 \Leftrightarrow g(x) > 0.5$.

- The $\ell_2$ norm certified radius for $g$ at $x_0$ is given as $R = \dfrac{\left|w^T x_0 + b\right|}{\|w\|_2}$.

We saw in subsection 3.2, the certified safety region calculated using only the zeroth order information has spherical symmetry. Thus, the certified safety region for $g$ at $x_0$ calculated using existing methods is a sphere of radius $R = \dfrac{\left|w^T x_0 + b\right|}{\|w\|_2}$ centered at $x$. For the proposed method we show using both the zeroth and the first order information gives :

**Proposition 3.** *Under our proposed method, the certified safety region for $g$ at a point $x_0$ is given as the halfspace $H = \{x \mid sign(w^T x + b) = sign(w^T x_0 + b)\}$.*

*Proof.* In this case, we can calculate

$$y^{(0)} = \Phi\left(\frac{\left|w^T x_0 + b\right|}{\sigma \|w\|_2}\right)$$

$$y^{(1)} = \frac{1}{\sqrt{2\pi\sigma^2}} e^{-\frac{(w^T x_0 + b)^2}{2\sigma^2 \|w\|_2^2}} \frac{sign(w^T x_0 + b)w}{\|w\|_2}$$

We shift the origin to $x_0$ and rotate and scale the basis by $\frac{1}{\sigma}$ to get a basis with positive $x_1$ along $sign(w^T x_0 + b)w$. Then, we can use the framework and to calculate the feasible set of $g'$'s. Any valid $g'$ can be written as $g' = f' \star \mu$ where

$$\int_{\mathbb{R}^d} f'(x) \left(\frac{1}{2\pi\sigma^2}\right)^{d/2} e^{-\frac{\|x\|_2^2}{2}} dx = p = y^{(0)}$$

$$\int_{\mathbb{R}^d} f'(x) \left(\frac{1}{2\pi\sigma^2}\right)^{d/2} x_1 e^{-\frac{\|x\|_2^2}{2}} dx = m = \sigma \left\|y^{(1)}\right\|_2 = \frac{1}{\sqrt{2\pi}} e^{-\frac{(\Phi^{-1}(1-p))^2}{2}}$$

Let $c = \Phi^{-1}(1 - p)$ and $f_0 = \mathbf{1}_{x_1 > c}$. It is easy to check that $f_0$ satisfies the above-mentioned constraints. We show that any $f'$ that satisfies the two constraints equals to $f$ almost everywhere. This is equivalent to saying $f_0 - f'$ is 0 almost everywhere. We see

$$\int_{\mathbb{R}^d} (f'(x) - f_0(x)) \left(\frac{1}{2\pi\sigma^2}\right)^{d/2} e^{-\frac{\|x\|_2^2}{2}} dx = 0$$

$$\int_{\mathbb{R}^d} (f'(x) - f_0(x)) \left(\frac{1}{2\pi\sigma^2}\right)^{d/2} x_1 e^{-\frac{\|x\|_2^2}{2}} dx = 0$$

Moreover, we have that for $x_1 > c$, $f'(x) - f_0(x) \leq 0$ and for $x_1 \leq c$, $f'(x) - f_0(x) \geq 0$. Thus, we can rewrite the first constraint as

$$\int_{x_1 > c} |f'(x) - f_0(x)| d\mu(x) = \int_{x_1 \leq c} |f'(x) - f_0(x)| d\mu(x)$$

For brevity we replaced the Gaussian integral over Lebesgue measure with an integral over the Gaussian measure. Now, for the second constraint we can re-write it as

$$\int_{\mathbb{R}^d} (f'(x) - f_0(x)) x_1 d\mu(x) = \int_{x_1 \leq c} |f'(x) - f_0(x)| x_1 d\mu(x) - \int_{x_1 > c} |f'(x) - f_0(x)| x_1 d\mu(x)$$

$$\leq c \int_{x_1 \leq c} |f'(x) - f_0(x)| d\mu(x) - c \int_{x_1 > c} |f'(x) - f_0(x)| d\mu(x)$$

$$= c \left( \int_{x_1 \leq c} |f'(x) - f_0(x)| d\mu(x) - \int_{x_1 > c} |f'(x) - f_0(x)| d\mu(x) \right)$$

$$= 0$$

$$\implies 0 = \int_{\mathbb{R}^d} (f'(x) - f_0(x)) x_1 d\mu(x) \leq 0$$

Thus the equality must hold in all the equations. Thus $\int_{x_1 > c} |f'(x) - f_0(x)| x_1 d\mu(x) = c \int_{x_1 > c} |f'(x) - f_0(x)| d\mu(x)$ which means $\int_{x_1 > c} |f'(x) - f_0(x)| d\mu(x) = 0$. Then using the results from the first constraint $\int_{x_1 \leq c} |f'(x) - f_0(x)| d\mu(x) = 0$. Thus,

$$\int_{\mathbb{R}^d} |f'(x) - f_0(x)| d\mu(x) = 0$$

As a result, $f'$ is equal to $f(0)$ almost everywhere w.r.t the Gaussian measure $\mu$. Thus, $g' = f' \star \mu = f_0 \star \mu$. Thus, we have only one feasible solution for $g'$ which is $g$. Thus, forall $x \in \mathbb{R}^d$ $\mathbf{p}_x(x) = g(x)$ and the certified safety region

$$\mathbf{SR}(x_0) = \{x \mid x \in \mathbb{R}^d; \ \mathbf{p}_x(x) > 0.5\} = \{x \mid x \in \mathbb{R}^d; \ g(x) > 0.5\}$$

Finally using the form of $g$ from [5, Appendix B]

$$\mathbf{SR}(x_0) = \{x \mid sign(w^T x + b) = sign(w^T x_0 + b)\}$$

$\square$

## C.1 Discussion

Using the result from Proposition 3 we see that using both zeroth and first order information allows us to give the optimal certified safety region for binary linear classifiers.

Although the results from zeroth order information give us the optimal $\ell_2$ radius as seen in Figure 3, the radius for other threat models like $\ell_1, \ell_\infty$ can be sub-optimal. Using additional first-order information allows us to overcome this problem. As seen in Figure 3, the safety region we achieve using the proposed work provides optimal radius for all $\ell_1, \ell_2, \ell_\infty$ threat models.

## D   Proofs for Numerical Estimation of First-Order Information

**Theorem 4.** *Given a black-box classifier $f$ and the random vector $z = w(f(x+w)_c - \frac{1}{2})$ where $w \sim \mathcal{N}(0, \sigma^2 \mathbf{I})$, we have that $z - \sigma^2 y^{(1)}$ is a sub-gaussian random vector with parameter $k = \sigma^2(\frac{1}{4} + \frac{3}{\sqrt{8\pi e}})$. For convenience, we do some abuse of notation to denote this as $(z - \sigma^2 y^{(1)}) \sim \mathrm{subG}(k)$.*

*Proof.* For any unit norm vector $v$ consider the moment generating function for the variable

$$\mathbb{E}[e^{sv^T(z-\sigma^2 y^{(1)})}] = e^{-sv^T \sigma^2 y^{(1)}} \mathbb{E}[e^{sv^T z}]$$

As the black-box classifier has binary output for every class, i.e, $f(x+w)_c$ is either 0 or 1, we have
$e^{sv^T z} = (e^{\frac{sv^T w}{2}} - e^{\frac{-sv^T w}{2}})f(x+w)_c + e^{\frac{-sv^T w}{2}}$

$$\mathbb{E}[e^{sv^T(z-\sigma^2 y^{(1)})}] = e^{-s\sigma^2 v^T y^{(1)}} \mathbb{E}[(e^{\frac{sv^T w}{2}} - e^{\frac{-sv^T w}{2}})f(x+w) + e^{\frac{-sv^T w}{2}}]$$
$$= e^{\frac{s^2 \sigma^2}{8} - s\sigma^2 v^T y^{(1)}} (1 + e^{\frac{-s^2 \sigma^2}{8}} \mathbb{E}[(e^{\frac{sv^T w}{2}} - e^{\frac{-sv^T w}{2}})f(x+w)])$$

Using Generalized Neymann-Pearson Lemma with the condition $\mathbb{E}[v^T w f(x+w)] = \sigma^2 v^T y^{(1)}$, we have

$$e^{-\frac{s^2 \sigma^2}{8}} \mathbb{E}[(e^{\frac{sv^T w}{2}} - e^{\frac{-sv^T w}{2}})f(x+w)] \leq \frac{1}{\sqrt{2\pi\sigma^2}} \int_{\frac{-\sigma^2|s|}{2}}^{\frac{\sigma^2|s|}{2}} e^{-\frac{(-\beta+z)^2}{2\sigma^2}} + e^{-\frac{(\beta+z)^2}{2\sigma^2}} - e^{-\frac{z^2}{2\sigma^2}} dz$$

where $s\sigma^2 v^T y^{(1)} = \frac{\sigma^2|s|}{\sqrt{2\pi\sigma^2}}(e^{-\frac{(-\beta)^2}{2\sigma^2}} + e^{-\frac{\beta^2}{2\sigma^2}} - e^{-\frac{0^2}{2\sigma^2}})$.

Let $\phi(w) = \frac{1}{\sqrt{2\pi\sigma^2}} e^{-\frac{(-\beta+w)^2}{2\sigma^2}} + e^{-\frac{(\beta+w)^2}{2\sigma^2}} - e^{-\frac{w^2}{2\sigma^2}}$. Then we have

$$\mathbb{E}[e^{sv^T(z-\sigma^2 y^{(1)})}] = e^{\frac{s^2 \sigma^2}{8} - sv^T \sigma^2 y^{(1)}} (1 + e^{\frac{-s^2 \sigma^2}{8}} \mathbb{E}[(e^{\frac{sv^T w}{2}} - e^{\frac{-sv^T w}{2}})f(x+w)])$$
$$\leq e^{\frac{s^2 \sigma^2}{8} - \sigma^2|s|\phi(0)} \left(1 + \int_{\frac{-\sigma^2|s|}{2}}^{\frac{\sigma^2|s|}{2}} \phi(w)dw\right)$$
$$\leq e^{\frac{s^2 \sigma^2}{8}} e^{\int_{\frac{-\sigma^2|s|}{2}}^{\frac{\sigma^2|s|}{2}} (\phi(w)-\phi(0))dw}$$

We see that the global Lipschitz constant for $\phi(w)$ is given as $\sup\|\phi'(w)\| \leq 3\sup\left\|\frac{w}{\sigma^2}\frac{1}{\sqrt{2\pi\sigma^2}} e^{\frac{-w^2}{2\sigma^2}}\right\| = \frac{3}{\sigma^2\sqrt{2\pi e}}$. Then we see that $\int_{\frac{-\sigma^2|s|}{2}}^{\frac{\sigma^2|s|}{2}} (\phi(w) - \phi(0))dw \leq \frac{3}{\sigma^2\sqrt{2\pi e}} \int_{\frac{-\sigma^2|s|}{2}}^{\frac{\sigma^2|s|}{2}} |w|dw = \frac{3s^2\sigma^2}{4\sqrt{2\pi e}}$. Thus.

$$\mathbb{E}[e^{sv^T(z-\sigma^2 y^{(1)})}] \leq e^{\frac{s^2 \sigma^2}{8}} e^{\frac{3s^2}{4\sqrt{2\pi e}}}$$
$$= e^{\frac{s^2}{2}\sigma^2(\frac{1}{4} + \frac{3}{\sqrt{8\pi e}})}$$

$\square$

**Corollary 4.1.** *For any $\alpha$, let $Z_n$ be the empirical mean of $n$ samples of the random variable $z$, then given $t_1 = \sqrt{\frac{2kd(d\log 2 - \log \alpha)}{n}}$, $t_\infty = \sqrt{\frac{2k(\log 2d - \log \alpha)}{n}}$*

$$\mathbb{P}\left(\left|\left\|y^{(1)}\right\|_1 - \|Z_n\|_1\right| \leq t_1\right) \geq 1 - \alpha, \quad \mathbb{P}\left(\left|\left\|y^{(1)}\right\|_\infty - \|Z_n\|_\infty\right| \leq t_2\right) \geq 1 - \alpha$$

*Proof.* Using Theorem 4 and the properties of subgaussian random vectors, we see that $Z_n \sim \mathrm{subG}(\frac{k}{n})$. Let the set of vectors $S = \{v \mid v \in \mathbb{R}^d; |v_i| = 1\}$, then $\|x\|_1 = \max_{v \in S} v^T x$. Using the

maximal property of sub-gaussian random variables over the set of $2^d$ variables $\{v^t Z_n \mid v \in S\}$ we get,

$$\mathbb{P}\left(\left\|y^{(1)} - Z_n\right\|_1 \leq t_1\right) \geq 1 - \alpha$$

By the triangle inequality $\left|\left\|y^{(1)}\right\|_1 - \|Z_n\|_1\right| \leq \left\|y^{(1)} - Z_n\right\|_1$ we get the first result. Similarly, $\|x\|_\infty = \max_i \max(e_i^T x, -e_i^T x)$ and once again using the maximal property of sub-gaussian random variables over the set of $2d$ variables $\{e_i^T Z_n, -e_i^T Z_n \mid e_i$ is a basis vector$\}$ we get,

$$\mathbb{P}\left(\left\|y^{(1)} - Z_n\right\|_\infty \leq t_\infty\right) \geq 1 - \alpha$$

Again using triangle inequality, we see $\left|\left\|y^{(1)}\right\|_\infty - \|Z_n\|_\infty\right| \leq \left\|y^{(1)} - Z_n\right\|_\infty$ proving the second inequality.

$\square$

**Lemma 4.** *If we have two sub-gaussian random vectors $X \sim \mathrm{subG}(k_1), Y \sim \mathrm{subG}(k_2)$ then*

$$\mathbb{P}(X^T Y < -t) \leq \max\left(e^{-\frac{t^2}{\sqrt{2}dk_1 k_2}}, e^{-\frac{t}{4\sqrt{k_1 k_2}}}\right), \quad \mathbb{P}(X^T Y > t) \leq \max\left(e^{-\frac{t^2}{\sqrt{2}dk_1 k_2}}, e^{-\frac{t}{4\sqrt{k_1 k_2}}}\right)$$

*Proof.* Consider the moment generating function for the variable $X^T Y$. We have for $|s| \leq \sqrt{\frac{2}{k_1 k_2}}$

$$\mathbb{E}[e^{sX^T Y}] \leq \mathbb{E}[e^{\frac{k_2 s^2 \|X\|^2}{2}}]$$

$$= \mathbb{E}[\mathbb{E}_{a\sim\mathcal{N}(0,r)}[e^{a^T X}]], \quad r = \frac{k_2 s^2}{2}$$

$$= \mathbb{E}_{a\sim\mathcal{N}(0,r)}[\mathbb{E}[e^{a^T X}]] \leq \mathbb{E}_{a\sim\mathcal{N}(0,r)}[e^{\frac{k_1 \|a\|^2}{2}}]$$

$$= \left(\frac{\frac{1}{\frac{1}{r}-k_1}}{r}\right)^{\frac{d}{2}} = \left(1 - \frac{s^2 k_1 k_2}{2}\right)^{\frac{-d}{2}}$$

Now we see that

$$\mathbb{P}(X^T Y < -t) = \mathbb{P}(e^{-sX^T Y} < e^{st}), s \geq 0$$

$$\leq \frac{\mathbb{E}[e^{-sX^T Y}]}{e^{st}}$$

$$\leq \left(1 - \frac{s^2 k_1 k_2}{2}\right)^{\frac{-d}{2}} e^{-st}$$

Taking $s = \sqrt{\frac{d^2}{4t^2} + \frac{2}{k_1 k_2}} - \frac{d}{2t}$ we see

$$\mathbb{P}(X^T Y < -t) \leq \left(1 + \frac{\sqrt{\frac{d^2}{4t^2} + \frac{2}{k_1 k_2}} - \frac{d}{2t}}{2\frac{d}{2t}}\right)^{\frac{d}{2}} e^{-\left(\sqrt{\frac{d^2}{4t^2} + \frac{2}{k_1 k_2}} - \frac{d}{2t}\right)t}$$

$$\leq e^{\frac{d}{2}\frac{\sqrt{\frac{d^2}{4t^2} + \frac{2}{k_1 k_2}} - \frac{d}{2t}}{2\frac{d}{2t}}} e^{-\left(\sqrt{\frac{d^2}{4t^2} + \frac{2}{k_1 k_2}} - \frac{d}{2t}\right)t}$$

$$= e^{-\left(\sqrt{\frac{d^2}{4t^2} + \frac{2}{k_1 k_2}} - \frac{d}{2t}\right)\frac{t}{2}} = e^{-\left(\sqrt{\frac{d^2 k_1 k_2}{8t^2}+1} - \frac{d\sqrt{k_1 k_2}}{\sqrt{8}t}\right)\frac{t}{\sqrt{2k_1 k_2}}}$$

$$\leq e^{-\frac{t}{\sqrt{2k_1 k_2}} \min\left(\frac{t}{d\sqrt{k_1 k_2}}, \frac{1}{\sqrt{8}}\right)}$$

$$\mathbb{P}(X^T Y < -t) \leq \max\left(e^{-\frac{t^2}{\sqrt{2}dk_1 k_2}}, e^{-\frac{t}{4\sqrt{k_1 k_2}}}\right)$$

We can use a similar proof to show

$$\mathbb{P}(X^T Y > t) \leq \max\left(e^{-\frac{t^2}{\sqrt{2}dk_1 k_2}}, e^{-\frac{t}{4\sqrt{k_1 k_2}}}\right)$$

$\square$

**Theorem 5.** *For any $\alpha \geq 2e^{-\frac{d}{16}}$, if we have two random vectors $X, Y$ such that $(X - \beta) \sim$ subG$(k_1)$ and $(Y - \beta) \sim$ subG$(k_2)$ then we can show that using $t = \sqrt{-\sqrt{2}k_1 k_2 d \log \frac{\alpha}{2}}$, $\epsilon_u = \sqrt{\frac{-(k_1+k_2)\log \frac{\alpha}{2}}{2(X^T Y + t)}}$, $\epsilon_l = \sqrt{\frac{-(k_1+k_2)\log \frac{\alpha}{2}}{2(X^T Y - t)}}$,*

$$\mathbb{P}\left( \|\beta\|_2 \leq \frac{\sqrt{X^T Y + t}}{\sqrt{1 + \epsilon_u^2} - \epsilon_u} \right) \geq 1 - \alpha, \quad \mathbb{P}\left( \|\beta\|_2 \geq \frac{\sqrt{X^T Y - t}}{\sqrt{1 + \epsilon_l^2} + \epsilon_l} \right) \geq 1 - \alpha$$

*Proof.* Using Lemma 4, we see $\mathbb{P}\left( X^T Y - \|\beta\|_2^2 \leq -t + \beta \cdot ((X - \beta) + (Y - \beta)) \right) \leq \max \left( e^{-\frac{t^2}{\sqrt{2}k_1 k_2 d}}, e^{-\frac{t}{4\sqrt{k_1 k_2}}} \right)$. Taking $t = \sqrt{-\sqrt{2}k_1 k_2 d \log \frac{\alpha}{2}}$, we see that for $\alpha \geq 2e^{-\frac{d}{16}}$ the first term is bigger. So

$$\mathbb{P}\left( \|\beta\|_2^2 \leq X^T Y + t - \beta \cdot (X + Y - 2\beta) \right) \geq 1 - \frac{\alpha}{2}$$

From the sub-gaussian property of $X, Y$, we have $\mathbb{P}(\beta \cdot (X + Y - 2\beta) \leq -t_1) \leq e^{-\frac{t_1^2}{2(k_1+k_2)\|\beta\|_2^2}}$. Taking $\epsilon_u = \sqrt{\frac{-(k_1+k_2)\log \frac{\alpha}{2}}{2(X^T Y + t)}}$ and $t_1 = 2\epsilon_u \|\beta\|_2 \sqrt{X^T Y}$, we get that

$$\mathbb{P}\left( X^T Y + t - \beta \cdot (X + Y - 2\beta) \leq X^T Y + t + 2\epsilon_u \|\beta\|_2 \sqrt{X^T Y + t} \right) \geq 1 - \frac{\alpha}{2}$$

Taking a union bound and combining the two inequalities we get

$$\mathbb{P}\left( \|\beta\|_2^2 \leq X^T Y + t + 2\epsilon_u \|\beta\|_2 \sqrt{X^T Y + t} \right) \qquad \geq 1 - \alpha$$

$$\iff \mathbb{P}\left( (1 + \epsilon_u^2)\|\beta\|_2^2 \leq \left( \sqrt{X^T Y + t} + \epsilon_u \|\beta\|_2 \right)^2 \right) \qquad \geq 1 - \alpha \qquad (10)$$

$$\iff \mathbb{P}\left( \|\beta\|_2 \leq \frac{\sqrt{X^T Y + t}}{\sqrt{1 + \epsilon_u^2} - \epsilon_u} \right) \qquad \geq 1 - \alpha$$

Using a similar proof we can also show that for $\epsilon_l = \sqrt{\frac{-(k_1+k_2)\log \frac{\alpha}{2}}{2(X^T Y - t)}}$, we have

$$\mathbb{P}\left( \|\beta\|_2 \geq \frac{\sqrt{X^T Y - t}}{\sqrt{1 + \epsilon_l^2} + \epsilon_l} \right) \geq 1 - \alpha$$

$\square$

Let $X = X_{n_1}, Y = Y_{n_2}$ be the empirical average of $n_1, n_2$ independent samples of the random variable $z$.

**Corollary 5.1.** *For any $\alpha \geq 2e^{-\frac{d}{16}}$, given $t = \sqrt{-k^2 \frac{\sqrt{2}d}{n_1 n_2} \log \frac{\alpha}{2}}$, $\epsilon_u = \sqrt{\frac{-k(n_1+n_2)\log \frac{\alpha}{2}}{2n_1 n_2 (X_{n_1}^T Y_{n_2} + t)}}$, $\epsilon_l = \sqrt{\frac{-k(n_1+n_2)\log \frac{\alpha}{2}}{2n_1 n_2 (X_{n_1}^T Y_{n_2} - t)}}$,*

$$\mathbb{P}\left( \left\| y^{(1)} \right\|_2 \leq \frac{\sqrt{X_{n_1}^T Y_{n_2} + t}}{\sqrt{1 + \epsilon_u^2} - \epsilon_u} \right) \geq 1 - \alpha, \quad \mathbb{P}\left( \left\| y^{(1)} \right\|_2 \geq \frac{\sqrt{X_{n_1}^T Y_{n_2} - t}}{\sqrt{1 + \epsilon_l^2} + \epsilon_l} \right) \geq 1 - \alpha$$

*Proof.* Using Theorem 4 and the properties of subgaussian random vectors, we see that $X_{n_1} \sim$ subG$(\frac{k}{n_1})$, $Y_{n_2} \sim$ subG$(\frac{k}{n_2})$. Then, using Theorem 5 we get the required values of $t, \epsilon_u, \epsilon_l$. $\square$

# E  Additional Experiments

Here, we give additional experiments on the Imagenet dataset. We reuse the models given by Cohen et al. [5] and calculate the certified accuracy at radius $R$ by counting the samples of the test set that are correctly classified by the smoothed classifier $g$ with certified radii of at least $R$. For both our proposed certificate and the baseline certificate [5], we use a failure probability of $\alpha = 0.001$ and $N = 200,000$ samples for CIFAR and $N = 1,250,000$ samples for Imagenet.

In the following plots we give a more detailed account of the improvement seen by using both the first and zeroth order information. For every trained model (depending on variance $\sigma$ used during training), we give the certified accuracy under $\ell_2$ norm threat model, $\ell_1$ norm threat model and the subspace $\ell_2$ norm threat model.

Figure 4: Certified Accuracy for Imagenet seen under various threat models and $\sigma$ values. The scale of $x$-axis is different for the 3 different models (denoted by training noise variance) as the certified radii we get for these three models have different ranges.

The findings here are similar to the ones reported for CIFAR. As expected we see from Figure 4 that the smallest improvements are for $\ell_2$ norm threat model where the new framework gives only marginal improvement over the $\ell_2$ radius certified by existing methods. This follows from the fact that the existing methods already produce near-optimal certified $\ell_2$ radii. However, certifying a significantly bigger certified safety region allows us to give significant improvements over the certified $\ell_1$ radius and the subspace $\ell_2$ radii (the subspace considered here is the red channel of the image, i.e., we only allow perturbations over red component of the RGB pixels of the image).

From these figures we are also able to see that, for any given model, most of the improvement in certified accuracy occurs at smaller values of radius $R$. We think one of the causes for this is the interval size of our estimates of $y^{(0)}, y^{(1)}$. So, we give the following results using larger number of samples to estimate both $y^{(0)}, y^{(1)}$.

In Figure 5 we show the results by using the observed values from experiments using $N_{obs} = 1250000$ and then constructing the estimates assuming we used $N = 200000, 800000, 1250000, 1600000, 3200000, 6400000$ samples respectively. Using these estimates we get a certified $\ell_1$ radius for both the existing method and for the proposed method. We see that

Figure 5: Effect of number of samples used on certified accuracy.

using larger number of samples allows us to get improvements at even larger values of $R$. However, we note that it is still not possible to get improvements at very high values of $R$. We think this would require very precise bounds for $y^{(0)}, y^{(1)}$ and thus a very high number of samples.