[Reviews · NeurIPS 2020]

Review 1

Summary and Contributions: This paper proposes to utilize the first-order information of the base function to improve the certified radius of randomized smoothing.

Strengths: The intuition of this paper is quite interesting. The black-box property of randomized smoothing is its advantage but also its weakness as it does not sufficiently exploit the property of the base function. Therefore, introducing its first-order information should be able to improve certified robustness. Although the proposed approach does not give a better robust radius on the l2 norm, which is expected as the bound from Cohen et al. is optimal, it can give a higher l1 radius for Gaussian smoothing. Utilizing first-order information in randomized smoothing is a difficult task, and the proposed solution to this problem is technically novel.

Weaknesses: 1. My major concern is that the improvement in l1 radius for Gaussian smoothing is somewhat limited. This paper also obtains better results for subspace perturbation, but it is not a common scenario in practice. I understand that it is important to consider the case where one model is used to defend against multiple types of threads, but this result might not be strong enough. 2. One thing that is not clear to me is how the l1 radius is calculated for Cohen et al. in Figure 2. In their paper, there is no mention of the l1 radius at all. Did the authors use bounds from other papers? It should be made clear. 3. Minor: the authors used "higher-order" in their paper but only discussed first-order in detail.

Correctness: I did not check all the proofs, but the key theorems are correct.

Clarity: The writing of this paper is clear and easy to follow.

Relation to Prior Work: The relationship to previous works is clearly discussed.

Reproducibility: Yes

Additional Feedback: I have read the rebutall and will keep my score unchanged. Thank you!


Review 2

Summary and Contributions: The authors proposed a new method to improve the robustness certificates for randomized smoothing-based classifiers. For black-box setting, their results show better performance.

Strengths: very detailed proof

Weaknesses: Not much experiment results was presented.

Correctness: maybe, I'm not familiar with this field of study.

Clarity: yes

Relation to Prior Work: yes

Reproducibility: No

Additional Feedback:


Review 3

Summary and Contributions: The authors propose a novel framework to incorporate higher order information regarding the smoothed classifier to break a fundamental limitation of certified robustness methods when dealing with lp norm attacks with large p. The main advantage of the framework is that the certified lp radius remains agnostic to the smoothing method. The main technical idea is to formulate the safety region around an input x as the intersection of safe regions of all smoothed classifiers that satisfy a certain condition, which also holds for the given smoothed classifier. This condition is originally only about the confidence of the most probable class in the smoothed classifier. The authors extend this constraint to be about the n-th order derivative of the smoothed function. For this to work, one has to guarantee the existence of such derivatives, which is done for a class of smoothing measures specified in the theorem 1. The proposed framework shows significant improvement of the l1 certified radius when smoothing is done with the Gaussian noise. It also shows marginal improvement for subspace l2 certified radius.

Strengths: - The paper is written well and clearly. - The framework breaks the impossibility result that certification can not be done for large p in lp norms. - The lp certification is also agnostic to the smoothing method.

Weaknesses: - The results for lp with larger p and l_inf are not given. It would be instructive to test against those cases and empirically demonstrate the full capacity of the method.

Correctness: Yes.

Clarity: Yes.

Relation to Prior Work: The related works are surveyed well enough.

Reproducibility: Yes

Additional Feedback:


Review 4

Summary and Contributions: In this paper, a general randomized smoothing framework is proposed to provide certified robustness against adversarial evasion attack. Compared to the previous randomized smooth paper published by Cohen et al in 2019, the contribution of this paper can be summarized as: 1. It introduces both the zero-th order and the first-order information of the classifier around a given input data x to shape the certified robustness area. Instead of using only isotropic Gaussian distribution for the smoothing use, integrating the first-order information makes the smoothing noise become directional, which is better adapted to the geometrical property of the local area around the input data x. 2. It proposes a theoretical framework that can be used to calculate the radius of the certified robust region measured by L1, L2 and Lp distance. It can be also adapted to the radius estimation in a subspace of the original feature space.

Strengths: 1. It provides a sounding theoretical analysis system. The detailed analysis show how to use both the zeroth order and the first-order structure of a classifier around one given data point x to estimate the shape of the certified robust area. It unveils the impactful factors that determine the non-isotropic distribution of the certified robust area, as shown by Theorem.3 and Proposition 1 and 2. For evasion attack, it is a well known observation that robustness / vulnerability of a classifier facing the attack depends on the local regularity of a classifier around a given input data point. This work shows in depth how to integrate the local regularity to describe the shape of the certified robust area and harden the classifier accordingly in a further step. 2. The proposed radius estimation method can be adapted to evasion attack scenarios with different distance metric (L1, L2 and Lp) enforced on attack budget. Furthermore, the proposed method can be used for hardening different classifiers, no matter whether they have linear or more complex structure. Only the first order information is required. It therefore can be applied as an universal defense strategy or attackability assessment method. In general, the topic discussed in this paper is very relevant to the NeurIPS community and the study is very interesting and innovative.

Weaknesses: 1. The presentation of the proof system is not clear enough, more clarification would be necessary: 1) In the discussion (Line 129), it is said “a smaller value p_{x}(z) for any x,z can make the super-level set of p_{x}(z) larger”. Intuitively, if p_{x}(z) becomes smaller, it means a small perturbation \delta will make the output of the classifier g biased from the expected class c, as only p_{x}(z) might be less than 0.5. So would smaller p_{x}(z) shrink the certified robust area ? 2) In Theorem.1, what is D^{alpha}_{x} ? Does it denote the derivatives of the probabilistic measure \mu? 3) Claim iii (Line 194) can not obviously be deduced from Figure.1. From the figure only it is difficult to justify “the directional certified radius is highest along the direction of the y^(1) and gets successively lower as we rotate away from it”. 4) Corollary 3.1 (Line 211) what are the variable z_{1} and z_{2}? Similarly, from Line 515 to Line 518 of the supplementary file, the optimization problem is reformulated by setting r = R/\sigma, but how is e^{Rv^{T}z/\sigma^2} changed to e^{rz_{1}} ? The set of {z_{i}} are widely employed in the proof system, it would be helpful to provide a clearer explanation to them. 2. Except from empirically showing better certified accuracy, the certified robust area can be also used to evaluate local robustness of a classifier around a given data point. Intuitively, if a data point stays far away from the classification boundary or the local structure of the classification boundary around a data point is smooth enough, it would be difficult to deliver an evasion attack with a limited attack budget. For the data point that is more difficult to be attacked, we would expect to observe that the certified robust area around this data point to be bigger than the others’. It would be interesting to verify empirically the existence of this association in reality. Furthermore,

Correctness: The empirical methodology is correct but would be great if it includes a study investigating the association between "how large the certified robust area of a data point is " and "how robust the classifier is around this data point". Overally the proof systems are well organized. But it would be necessary to make the definitions of the introduced terms (e.g. z_{i}) more clear.

Clarity: Yes

Relation to Prior Work: Yes, the difference between this work and the previous randomized smoothing method is discussed thoroughly.

Reproducibility: Yes

Additional Feedback: We have read through the feedbacks from the reviewers. We would like to keep our ratings unchanged.

[Author Response · NeurIPS 2020]

We thank reviewers for their thoughtful feedback. We are pleased to see that most reviewers found the work to be interesting and innovative. Here, we provide clarifications and conduct additional experiments to demonstrate the improvement in performance achieved by our proposed technique for certification under the $\ell_\infty$ norm threat model.

**Reviewer 1 : 1) Significance of results.** Although we only give empirical evidence for $\ell_1, \ell_2$ norm and subspace $\ell_2$ norm in the manuscript, the theoretical guarantees in fact extend to $\ell_\infty$ and subspace $\ell_1, \ell_\infty$ norms. We only focus on $\ell_1, \ell_2$ and $\ell_\infty$ as these are the most intensely researched / relevant threat models in the field. In response to Reviewer 3's comments, we also provide empirical evidence to show improvement for $\ell_\infty$ norm on the CIFAR10 dataset. As the current state-of-the-art for $\ell_\infty$ norm is given under Gaussian smoothing [1], our empirical result can give the new state-of-the-art for $\ell_\infty$ norm certification. **2) Cohen $\ell_1$ radius calculation :** The $\ell_1, \ell_\infty$ norm results were not explicitly stated in the original paper [2] but they can be derived by following the same analysis, which are also stated in the recent paper [1, Appendix Table A] . **3) "Higher-order" in title :** The "higher-order" in the title is in reference to the fact that the paper lays down the ground work for using higher-order information for certification. However, we see that it might be ambiguous as we only fully explore first-order smoothing. So, we plan to change the title to "first-order smoothing".

**Reviewer 2 : Limited empirical evidence :** We note that our experiments provide numerical evidence of our theoretical results and demonstrate that the certification performance can be greatly improved by incorporating higher-order information. We have followed standard experiment setup and conducted various experiments on CIFAR10 (Sec 5) and ImageNet (Appendix E) and compared all the current baselines the $\ell_1, \ell_2$ norm and subspace $\ell_2$ norm, which is in line with other works in this field. Notably, we conduct additional CIFAR10 experiments for $\ell_\infty$ certification in Figure R1.

**Reviewer 3 : Experiments for $\ell_\infty$ :** In the current paper, we have only focused on giving the bounds for $\ell_1, \ell_2, \ell_\infty$ norms. For general $\ell_p$ norm we can use the current results to provide lower bounds on the certified radii. As for empirical results for certifying the $\ell_\infty$ norm radius, it requires the estimation of $\left\| y^{(1)} \right\|_1$. As mentioned in line 270 in the paper the current estimators used to calculate $\left\| y^{(1)} \right\|_1$ need a lot of samples in order to find non-vacuous high-confidence bounds. Although the certification cost is higher, **using the proposed method gives us significant ($\sim 10\%$) improvement over the bounds given by Cohen et al. for CIFAR10.** The CIFAR10 $\ell_\infty$ results in Figure R1 are calculated using 4M samples for certification (6 minutes/image). One of the major limitations of the current estimators is that they are biased. In the paper we have proposed a new unbiased estimator for $\left\| y^{(1)} \right\|_2$ (Table 1 in the current paper). We think a similar new estimator is needed to make $\ell_\infty$ certification more scalable. This is left for future work.

Figure R1: Comparing certified accuracy for CIFAR10 under $\ell_\infty$ threat models. Our results show that around $10\%$ improvement can be obtained by using the proposed method.

**Reviewer 4 : 1) Typos and clarifications : *i)*** Sorry it is a major typo. The statement should read *"Under the proposed general framework for calculating certified radii, it is easy to see that adding more local constraints ($H_i^x$) in Equation (2) gives a ~~smaller~~ bigger value of $\boldsymbol{p}_x(z)$ for any $x, z$ which makes the super-level set of $\boldsymbol{p}_x$, equivalently the certified safety region, bigger." **ii)*** With a slight abuse of notation, we use $\mu$ to denote both the measure and the probability density function of the measure. Here, $D_x^\alpha \mu(y-x)$ corresponds to taking the multivariate differential of the probability density function ($\mu$) at $(y-x)$ with respect to variable $x$. ***iii)*** In Figure 1 of our manuscript, the direction of the gradient $y^{(1)}$ is along the negative x-axis. We plan to add an arrow to clarify this. Also in order to better motivate the idea we plan to add numerical values for the directional certified radii on the figure. ***iv)*** Given the images in the pixel space, we do change of basis to orient the basis along the gradient $y^{(1)}$ to simplify calculations. In line 515-518, $z_1, z_2, \ldots, z_d$ denotes the variables corresponding to the new basis vectors for the space after the transformation. In corollary 1 we abuse the notation and use $z_1, z_2$ to denote the variables involved in the system of equations we reduce our initial constraints to. The two sets of variables are not linked. We will change the variable names to avoid confusion in the future and also give a description of $z_i$'s in the proof before using them. **2) Certified bounds vs Attack bounds :** We do agree that the experimental evidence would be great. However we are aware of attacks on randomized smoothing classifiers only for the $\ell_2$ norm threat model currently. For this scenario the current bounds are near optimal as the attacks are close to the current state-of-the-art certification bounds [3] (equivalently our proposed certification bounds).

# References

[1] G. Yang, T. Duan, E. Hu, H. Salman, I. Razenshteyn, and J. Li, "Randomized smoothing of all shapes and sizes," *ICML*, 2020.

[2] J. M. Cohen, E. Rosenfeld, and J. Z. Kolter, "Certified adversarial robustness via randomized smoothing," *ICML*, 2019.

[3] H. Salman, J. Li, I. Razenshteyn, P. Zhang, H. Zhang, S. Bubeck, and G. Yang, "Provably robust deep learning via adversarially trained smoothed classifiers," *Neural Information Processing Systems*, 2019.


[Meta-Review · NeurIPS 2020]

Randomized smoothing is one of the most promising techniques for adversarial training with guarantees. This work further generalizes randomized techniques with new methods, overcoming previous impossibility results for randomized smoothing with certain norms. This work is likely to be very influential among people studying adversarial machine learning.